

# Sensitivity Analysis of Numerical Modeling Input Parameters on Floating Offshore Wind Turbine Loads

Will Wiley[1], Jason Jonkman[1], Amy Robertson[1], and Kelsey Shaler[1]

[1]National Renewable Energy Laboratory, 15013 Denver West Parkway, Golden, CO 80401, USA

**Correspondence:** Will Wiley (will.wiley@nrel.gov)

**Abstract.**

Floating wind turbines must withstand a unique and challenging set of loads from the wind and ocean environment. To derisk development, accurate predictions of these loads are necessary. Uncertainty in modeling predictions leads to larger required safety factors, increasing project costs and the levelized cost of energy. Complex aero-hydro-elastic modeling tools use many

input parameters to represent the wind, waves, current, aerodynamic loads, hydrodynamic loads, and structural properties. It is helpful to understand which of these parameters ultimately drives a design. In this work, an ultimate and fatigue loads sensitivity analysis was performed with 35 different input parameters, using an elementary effects approach to identify the most influential parameters for a case study involving the NREL 5-MW baseline wind turbine atop the OC4-DeepCwind semisubmersible during normal operation. The importance of each parameter was evaluated using 14 response quantities of

interest across three operational wind speed conditions.

The study concludes that turbulent wind velocity standard deviation is the parameter with the strongest sensitivity; this value is important not just for turbine loads, but also for the global system response. The system center of mass in the wind direction is found to have the highest impact on the system rotation and tower loads. The current velocity is found to be the most dominating parameter for the system global motion and consequently the mooring loads. All tested wind turbulence

parameters in addition to the standard deviation, are also found to be influential. Wave characteristics are influential for some fatigue loading, but did not significantly impact the extreme ultimate loads in these operational load cases.

## 1 Introduction

It is projected that over 8 GW of floating wind energy will be installed by 2027 (Musial et al., 2022). Floating offshore wind turbines (FOWTs) experience a unique set of loads and responses with complicated physics governing the design require-

ments. Modeling tools that provide accurate load predictions are required for safe and efficient development. The International Electrotechnical Commission (IEC) Technical Specification 61400-3-2 lays out simulations that should be run and safety criteria that need to be met (IEC, 2019a). The outputs from modeling tools are functions of a very large set of input parameters including the environmental excitations, system properties, and aerodynamic and hydrodynamic modeling coefficients.

Not all inputs have an equal impact on the FOWT loads, and different types of loads are more sensitive to different inputs.

Most modeling parameters have some uncertainty associated with them. This uncertainty could be due to imprecision in





defining the parameter, physical statistical variability in the parameter, or changes in time throughout the life of a project. The impact of this uncertainty should be assessed for uncertainty in the ultimate and fatigue loads on the structure. It is helpful to understand which modeling inputs are really driving the uncertainty, so the analysis can be focused and efficient.

Sensitivity studies have previously been done for land-based and offshore wind turbines. One approach to assess sensitivity
is the elementary effects (EE) method. This method has been commonly used in the wind industry, where variance based sensitivity approaches are difficult given the complexity of the modeling. The EE method is sometimes called a screening method, and can effectively identify the set of most influential parameters, but does not consider any coupling of inputs (Morris, 1991).

In 2018, Robertson et al. used EE to assess the significance of 18 turbulent wind-inflow related parameters for three different
wind speed ranges on a land-based NREL 5-MW baseline wind turbine. They found that shear and turbulence levels in the main wind direction were the most important parameters impacting the turbine loads (Robertson et al., 2018). They used uniform perturbations for each of the parameters and found that with enough sampled points, the results mirrored sensitivity approximations (Robertson et al., 2018). A 2019 study by Shaler et al. replicated this approach but focused fully on airfoil properties. It was found that the turbine loads were most sensitive to aerodynamic lift coefficient and blade twist distribution
(Shaler et al., 2019). Again, this study broke the simulations into three different wind speed ranges, where they found that the ultimate loads were strongly stratified by the wind condition, while the fatigue loads were more similar between conditions (Shaler et al., 2019). Another 2019 analysis used the same EE method to look not only at wind and aerodynamics, but also mass, structural, and control parameters with important uncertainties; this work again found that turbulence in the wind direction and shear were the most influential inputs, but also identified new sensitivities to turbine yaw misalignment and the lift distribution
on the outboard section of the blade (Robertson et al., 2019b).

Wind turbine simulations are very complex, with a huge number of highly coupled input parameters. The EE approach with independent radial perturbations has been well tested for effects at the turbine level in the papers cited above. This process was extended to the farm level, with an emphasis on wake effects, in a 2021 project (Shaler et al., 2021). The turbine interactions were evaluated using three turbines, and 28 wake and inflow related parameters were studied. Again, even when considering
multiple turbines, the turbulence in the wind direction and wind shear dominated the ultimate and fatigue loads (Shaler et al., 2021).

Offshore wind energy deployments introduce a wide range of additional input parameters for the ocean environment and the complex support structures. A 2023 study assessed the sensitivity of dynamic modal parameters to four modeling inputs: wind speed, rotor speed, nacelle yaw angle, and mean sea level, for the fixed bottom Block Island wind turbines (Partovi-Mehr et al.,
2023). The study compared multiple modeling tools including OpenFAST and validated with field data from the operating wind farm. The group found that for this jacket supported turbine, the system natural frequency was most strongly dependent on the rotor speed and the system damping ratio was most strongly dependent on the wind speed (Partovi-Mehr et al., 2023).

A 2022 study used the EE method to look at the sensitivity of fatigue to a broad set of continuous and discrete parameters, including cycles to failure curves for the tower and monopile (Sørum et al., 2022). The project studied three different monopile-
supported large offshore wind turbines to assess if certain sensitivities were turbine specific. They found that the cycles to





failure and fatigue capacity parameters had the largest influence (Sørum et al., 2022). Environmental variables had a secondary impact, and both wind and wave values were important; wind conditions drove the tower fatigue while wave conditions drove the monopile fatigue (Sørum et al., 2022). Across the three tested turbines, the wind was more influential compared to the waves for the turbines with the larger rated power (Sørum et al., 2022).

Floating platforms introduce more complexity. Not only are new modeling parameters involved, but the critical response can involve a new range of motions. It is predicted that FOWT costs can potentially be reduced more than three times by 2030 compared to 2021 costs (Musial et al., 2022). To achieve these drastic gains in efficiency and do so in a safe way, it is important that designers can assess the uncertainty in their loads predictions. The previously demonstrated EE method can help identify which modeling parameters have the greatest impact on FOWT loads, isolating the focus for in-depth uncertainty studies.

## 2 Approach

The EE method tested with previous sensitivity studies was extended to a semi-submersible floating wind platform. A range of input parameters was selected, focusing on inputs previously identified as having a strong sensitivity, and adding new offshore and floating support-structure-specific variables. The tested ranges of the input variables were chosen based on possible or expected levels of uncertainty. The study was conducted for an operating wind turbine in three different wind conditions: below-rated, near-rate, and above-rated. These conditions correspond to wind speeds of 8.0 m/s, 12.0 m/s, and 18.0 m/s respectively, and also included wave and current conditions. Some input parameter ranges are functions of the wind speeds while some are constant across the conditions.

### 2.1 Wind Turbine and Floating Platform

The subject wind turbine is the NREL offshore 5-MW baseline wind turbine. This device has open-source characteristics and has been used in many research efforts. It was designed to be representative of commercial turbines to help enable and advance conceptual design studies. The turbine features an upwind three-bladed rotor and uses variable speed and collective pitch control (Jonkman et al., 2009).

The platform used in the study is the OC4-DeepCWind semi-submersible designed for the DeepCWind Consortium and adapted for a collaborative verification and validation project (Robertson et al., 2014). This platform has publicly available properties. It was the subject of multiple phases of the International Energy Agency (IEA) Wind Task 30 Offshore Code Comparison Collaborative (OC4-OC6) project, and has generally well understood behavior (Robertson et al., 2014). The design features three offset columns with heave plate type caps at the bottom. A smaller diameter central column supporting the wind turbine tower is connected with a system of more slender pontoons and braces. Figure 1 shows the configuration used in the OC4 project, which also was paired with the NREL 5-MW turbine. The platform is held in place by a catenary mooring system with three radial lines connecting three anchors to fairleads near the base of the columns at the tops of the heave plates.





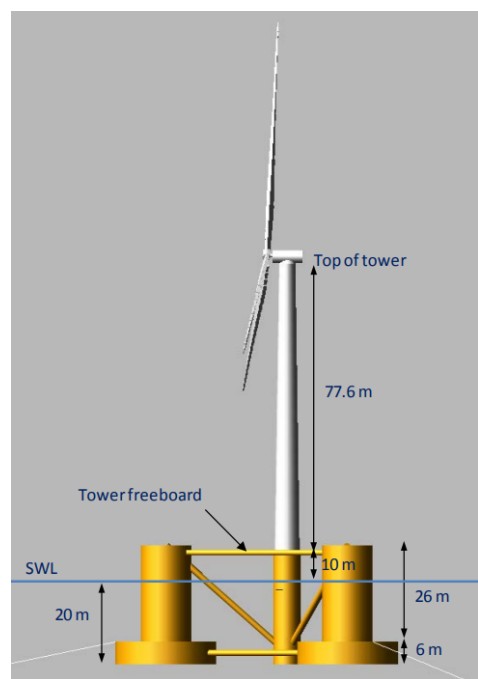

**Figure 1.** DeepCWind Floating Offshore Wind Platform (Robertson et al., 2014)

## 2.2   Modeling Approach

The numerical modeling was performed using OpenFAST version 3.3.0 for the aero-hydro-servo-elastic analysis. The turbulent inflow wind was generated using TurbSim. The hydrodynamic forces on the floating platform were calculated in the HydroDyn module using a hybrid combination of potential flow coefficients and viscous-drag elements. The irregular wave field was generated using the JONSWAP spectrum and the current was depth independent. Note that wave stretching was not included; this will lead to an underprediction of second order wave forces and a reduced sensitivity to the hydrodynamic forces near the waterline. The first- and second-order potential flow calculations were performed using the panel method code WAMIT. The mooring system was modeled using the lumped-mass-dynamics-based MoorDyn module. The substructure was treated as rigid, but the tower and blades were compliant, and their deflections were calculated using the ElastoDyn module. The aerodynamic forces were calculated in the AeroDyn module, using blade element momentum theory with unsteady aerodynamics and tower forces. The turbine was controlled using the ServoDyn module, using the baseline controller for the NREL 5-MW turbine atop the OC4-DeepCwind semisubmersible with a Bladed-style dynamic link library.

Each simulation was run for a 10-minute time series with a 1-minute transient removed from the results. IEC recommends using at least 6 10-minute simulations, with potentially more depending on the specific FOWT and site IEC (2019a). The IEC floating wind specification encourages the use of more random seed numbers instead of longer simulations (IEC, 2019a). In




this work, the simulation time is held constant and the number of seeds is increased until the results are no longer dependent on the number of seeds as described in Section 6.

## 2.3 Elementary Effects

The elementary effects method was outlined by Max Morris in 1991 for general computational experiments. The key advantage
is one factor is adjusted at a time, reducing the total number of simulations required for a sensitivity assessment (Morris, 1991). This approach does not identify coupling between input parameters, but is effective for screening for the variables creating the largest sensitivity.

A modification of the original technique uses radial perturbations of all parameters for a sufficiently large number of starting points. This radial EE method was described and tested by Robertsen et al. in 2018, and used here (Robertson et al., 2018).
The technique is graphically represented in Figure 2 for a simplified model with only three input parameters. Each blue dot represents a position in the parameter hyperspace, and the red dots represent a small shift from that point for one of the parameters. The effect on the output quantity due to the perturbation informs the sensitivity to that parameter at that specific location in the possible range of inputs. When each parameter is perturbed, their relative influence and importance can be determined. As the number of blue dots, or starting set of conditions, grows, the results converge towards the global sensitivity.
This process can be extended to any number of input parameters, and the results can be assessed for any number of output quantities of interest.

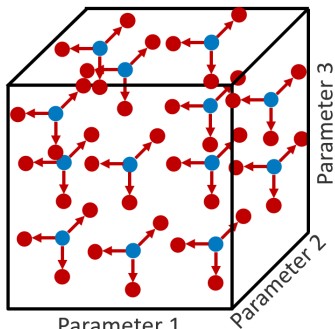

**Figure 2.** Three parameter representation of input parameter hyperspace with set of starting points and individual parameter perturbations, adapted from (Robertson et al., 2019b)

In this work, the size of the perturbations was held constant, to ±10% of the parameter range. The direction of the perturbation (plus or minus) was randomly chosen, with the constraint that the perturbation had to remain inside the possible range. The starting points were chosen following a Sobol sequence. This sequence is designed to have a uniform-type distribution in
an n-dimensioned space (Sobol, 1967). This approach avoids biasing the results to some location in the parameter ranges, and allows the addition of new starting points after simulations have been run.



Irregular wave conditions and turbulent wind conditions require a seed for the pseudo-random number generator (pRNG) for phase information. There can be variation in loads based only on the seed, so a sufficient number of seeds should be tested to ensure the influence in seed does not mask the influence of the parameter variation. The IEC recommends that at least six different seeds be run for 10-minute floating offshore wind simulations for each load case (IEC, 2019a). Their guidance states that this minimum may be higher depending on a the specific device and environmental condition (IEC, 2019a). In this work the same number of seeds were used at every starting point and perturbation. The sets of seeds were chosen to be unique for every starting point and perturbation. The necessary number of seeds was specifically considered for this model.

The output quantities of interest ($Y$) were combined for all seeds ($S$) of a certain set of inputs ($U$) following Equation 1 and 2, for the ultimate and fatigue loads respectively. The ultimate load ($Y_{ult.}$) is taken as the mean of the absolute maxima across each seed ($s$). The fatigue load ($Y_{fat.}$) is taken as the mean of the standard deviations across each seed. This evaluation is not a true representation of fatigue loads, as only the size of the load cycles is considered, not the number of cycles. It is expected that the relevant frequencies for each load output will be similar enough for the standard deviation to serve as a reasonable proxy for the fatigue. This was done to simplify the post-processing where detailed structural and material properties are not fully known. It is possible that if true fatigue calculations were performed, the relative fatigue sensitivity of input parameters highly impacting system frequencies could be larger.

$$Y_{ult.}(U) = \frac{1}{S} \sum_{s=1}^{S} MAX(|Y(U)|) \tag{1}$$

$$Y_{fat.}(U) = \frac{1}{S} \sum_{s=1}^{S} STD(Y(U)) \tag{2}$$

The EE sensitivity value for an input parameter ($i$) at a certain starting point in the parameter hyperspace ($b$) for a certain wind speed condition ($w$) is defined for ultimate loads in Equation 3 and for fatigue loads in Equation 4. Both equations feature a local partial derivative type calculation, with the ratio of the change in output to the change in input. $U^b$ is the set of all conditions at the starting point, and $U^b + x_i$ is the set of conditions including the perturbation in the $i$ input parameter. The partial derivative is multiplied by the total range for the $i$ input parameter, because it is not just that gradient that is important to sensitivity, but also the range. As a result, the sensitivity values from different input parameters can all be compared in the units of the output value. This results in simply multiplying the change in output by the inverse of the chosen $\Delta$ value, representing the variation in output across the full input range based on the local sensitivity.

There are likely large differences in the output loads based on the wind speed condition. For ultimate loads, the maximum value experienced is a key concern, so if sensitivity is high, but the total load is still low, it is not important. To account for this, the ultimate sensitivity value includes the addition of the output based on all input parameter nominal values corresponding to the wind speed ($Y_w$).

$$UEE_{iw}^b = \left| \frac{Y_{ult.}(U^b + x_i) - Y_{ult.}(U^b)}{\Delta_{iw}} u_{iw,range} \right| + \overline{Y_w} \tag{3}$$





For fatigue loads, if the sensitivity is high, but the number of relevant cycles are very low in the device's lifetime, it will not drive the design. The fatigue sensitivity value is thus scaled by the probability of occurrence of the corresponding wind speed condition ($P(w)$).

$$160 \quad FEE_{iw}^b = P(w) \left| \frac{Y_{fat.}(U^b + x_i) - Y_{fat.}(U^b)}{\Delta_{iw}} u_{iw,range} \right| \tag{4}$$

The probability of each wind condition was calculated based on an aggregate model of the United States offshore wind locations created in a 2016 study based on data from the National Data Buoy Center (Stewart et al., 2015). The project created comprehensive joint probability distributions for "wind speed, significant wave height, wave peak-spectral period, and wind/wave misalignment" for the West Coast, East Coast, and Gulf Coast (Stewart et al., 2015). The wind condition probabilities were based on a combination of the three representative sites with an even weighting, and are found in Table 1. The total of the probabilities does not equal 100% because the speeds below cut-in and above cut-out are not included in this study. This is not a problem for the weighting done in Equation 4.

**Table 1.** Wind condition probability used for fatigue elementary effects sensitivity calculation

| Condition | Below-Rated | Near-Rated | Above-Rated |
|---|---|---|---|
| **Wind Speed Range [m/s]** | 3.0 - 9.0 | 9.0 - 15.0 | 15.0 - 25.0 |
| **Aggregate Probability** | 0.525 | 0.322 | 0.062 |

To identify which input parameters contributed to the highest sensitivity, significant events were defined following Equations 5 and 6, where an EE value greater than two times the standard deviation larger than the mean EE value is counted. The number of times that an individual input parameter led to a significant EE value quantifies how sensitive the output is to that parameter relative to the rest of the studied variables.

$$UEE_{significant} > \overline{UEE} + 2\sigma_{UEE} \tag{5}$$

$$FEE_{significant} > \overline{FEE} + 2\sigma_{FEE} \tag{6}$$

## 3 Input Parameters

Table 2 lists all 35 studied input parameters. The variables are colored by five categories: red for wind, magenta for aerodynamics, blue for sea state, cyan for hydrodynamics, and black for system and structure. The first 16 parameters have been previously identified in elementary effects studies as having significant sensitivity (Robertson et al., 2018; Shaler et al., 2019; Robertson et al., 2019b). The last 19 parameters are additions for a floating offshore system that are expected to have potential



180   importance. Some critical parameters were highlighted in a 2019 analysis that studied hydrodynamic uncertainty for the OC5-DeepCWind semisubmersible, with a particular focus on numerical underprediction of low-frequency response (Robertson et al., 2019a). The mooring stiffness (dependent on mooring weight for a catenary system), the system center of gravity, and the wave amplitude were found to all contribute significant uncertainty (Robertson et al., 2019a). An important difference in this 2019 study is that an operating wind turbine was not present.



**Table 2.** Modeling input parameters for sensitivity analysis (red = inflow wind, magenta = aerodynamic forces, black = system and structure, blue = sea state, cyan = hydrodynamic forces)

| Input Parameter | Label | Units |
|---|---|---|
| Shear exponent | $Shear$ | - |
| Veer | $Veer$ | degrees |
| Coherence exponent | $\gamma_{wind}$ | - |
| σu (standard deviation) | $\sigma_u$ | m/s |
| Lu (integral scale parameter) | $L_u$ | m |
| au (coherence decrement) | $a_u$ | - |
| bu (offset parameter) | $b_u$ | 1/m |
| Yaw error | $Yaw$ | degrees |
| Air density | $\rho_{air}$ | kg/m$^3$ |
| Blade mass | $M_{blade}$ | % |
| Blade mass imbalance (factor) | $M_{blade}IMB$ | % |
| Blade twist at tip | $Twist$ | degrees |
| Airfoil lift coefficient at root | $Cl_{root}$ | % |
| Airfoil lift coefficient at tip | $Cl_{tip}$ | % |
| Tower stiffness factor | $TK$ | - |
| Tower damping tuner | $TD$ | - |
| Mooring mass per unit length | $M_{mooring}$ | kg/m |
| System center of mass - upstream/upwind | $SCM_X$ | m |
| System center of mass - vertical | $SCM_Z$ | m |
| System inertia - pitch direction | $SI_P$ | kg-m$^2$ |
| System inertia - yaw direction | $SI_Y$ | kg-m$^2$ |
| System mass | $M_{system}$ | kg |
| Water depth | $Depth$ | m |
| Water density | $\rho_{water}$ | kg/m$^3$ |
| Directional spreading range | $\Theta_{spread}$ | degrees |
| Wind/wave misalignment | $\Theta_{mis}$ | degrees |
| Significant wave height | $Hs$ | m |
| Peak wave period | $Tp$ | s |
| Wave spectral shape factor | $\gamma_{wave}$ | - |
| Current speed | $V_{current}$ | m/s |
| Current direction | $\Theta_{current}$ | degrees |
| Drag coefficient near water line | $Cd_{WL}$ | - |
| Drag coefficient on column | $Cd_{top}$ | - |
| Drag coefficient on heave plate | $Cd_{bottom}$ | - |
| Axial drag coefficient | $Cd_{axial}$ | - |





## 3.1 Parameter Ranges

For each input parameter, the potential range of uncertainty and a nominal value was selected. The ranges are either based on statistical variation in time or uncertainty. These values are shown in Table 3. The minimum, nominal, and maximum values are given for each of the three wind bins if there is a dependence (otherwise, the same value is used for all wind speed bins). The nominal values are used for the ultimate elementary effects sensitivity calculation as described in Equation 3, and the minimum and maximum values determine the range that the parameters can lie between.

In general the variables are independent of each other. The exception to this is coupling between wind and wave misalignment, significant wave height, and peak wave period. The maximum wave height is a function of the wind and wave misalignment; waves can potentially be larger when the misalignment is small. Both the minimum and maximum wave periods are functions of the wave height.



**Table 3.** Modeling input parameter ranges and nominal values for below rated (**BR**), near rated (**NR**), and above rated (**AR**) wind speed bins (red = inflow wind, magenta = aerodynamic forces, black = system and structure, blue = sea state, cyan = hydrodynamic forces) If only one value is given it applies to all 3 wind conditions

| Input Parameter | Units | | Minimum | | | Nominal | | | Maximum | |
|---|---|---|---|---|---|---|---|---|---|---|
| | | BR | NR | AR | BR | NR | AR | BR | NR | AR |
| Shear exponent | - | | -0.03 | 0.05 | 0.06 | | | 0.14 | 0.36 | 0.26 | 0.18 |
| Veer | degrees | | -7.6 | -6.688 | -6.688 | | | 0 | 24.776 | 28.728 | 28.728 |
| Coherence exponent | - | | | | 0 | | | 0 | | | 1 |
| $\sigma u$ (standard deviation) | m/s | | 0.28 | 0.42 | 0.92 | 0.952 | 1.23 | 1.796 | 1.62 | 2.04 | 2.67 |
| Lu (integral scale parameter) | m | | 5.0 | 8.0 | 25.0 | | | 340.2 | 1000.0 | 1400.0 | 1600.0 |
| au (coherence decrement) | - | | | | 1.5 | | | 12.0 | | | 26.0 |
| bu (offset parameter) | 1/m | | | | 0 | | | 0.00035 | 0.08 | 0.08 | 0.05 |
| Yaw error | degrees | | | | -20 | | | 0 | | | 20 |
| Air density | kg/m$^3$ | | | | 1.2005 | | | 1.225 | | | 1.2495 |
| Blade mass | % | | | | 0.993 | | | 1.045 | | | 1.098 |
| Blade mass imbalance (factor) | % | | | | 0 | | | 0 | | | 0.05 |
| Blade twist at tip | degrees | | | | -1.894 | | | 0.106 | | | 2.106 |
| Airfoil CL at root | % | | | | -0.26 | | | 0 | | | 0.26 |
| Airfoil CL at tip | % | | | | -0.26 | | | 0 | | | 0.26 |
| Tower stiffness factor | - | | | | 0.7225 | | | 1 | | | 1.3225 |
| Tower damping tuner | - | | | | 0.001 | | | 0.0255 | | | 0.05 |
| Mooring mass per unit length | kg/m | | | | 112.22 | | | 113.35 | | | 114.48 |
| System center of mass - upstream/upwind | m | Nacelle | | | 1.86 | | | 1.90 | | | 1.94 |
| | | Platform | | | -4.0 | | | 0.0 | | | 4.0 |
| System center of mass - vertical | m | Nacelle | | | 1.72 | | | 1.75 | | | 1.79 |
| | | Platform | | | -9.35 | | | -8.66 | | | -7.97 |
| System inertia - pitch direction | kg-m$^2$ | | | | 2.357E+09 | | | 2.562E+09 | | | 2.767E+09 |
| System inertia - yaw direction | kg-m$^2$ | Nacelle | | | 2.556E+06 | | | 2.608E+06 | | | 2.660E+06 |
| | | Platform | | | 3.903E+09 | | | 4.243E+09 | | | 4.582E+09 |
| System mass | kg | Nacelle | | | 2.352E+05 | | | 2.400E+05 | | | 2.448E+05 |
| | | Platform | | | 3.544E+06 | | | 3.852E+06 | | | 4.160E+06 |
| Water depth | m | | | | 196 | | | 200 | | | 204 |
| Water density | kg/m$^3$ | | | | 1020 | | | 1025 | | | 1030 |
| Directional spreading range | degrees | | | | 0 | | | 0 | | | 20 |
| Wind/wave misalignment | degrees | | | | 0 | | | 0 | | | 90 |
| Significant wave height | m | | | | 0 | 1.45 | 1.95 | 3.14 | | | $f(\Theta_{mis})$ |
| Peak wave period | s | | | | $f(H_s)$ | 8.25 | 7.96 | 8.42 | | | $f(H_s)$ |
| Wave spectral shape factor | - | | | | 1.0 | 1.0 | 1.0 | 0.284 | | | 7.0 |
| Current speed | m/s | | | | 0.0 | | | 0.0 | | | 2.0 |
| Current direction | degrees | | | | 0 | | | 0 | | | 90 |
| Drag coefficient at water line | - | | | | 0.4 | | | 1.2 | | | 2 |
| Drag coefficient on column | - | | | | 0.4 | | | 0.7 | | | 1 |
| Drag coefficient on heave plate | - | | | | 0.4 | | | 1.2 | | | 2 |
| Axial drag coefficient | - | | | | 3.5 | | | 4.8 | | | 5.5 |



The first 16 parameter ranges were based on the ranges from previous elementary effects studies with some adjustments for an offshore wind environment (Robertson et al., 2019b; Shaler et al., 2019, 2021). The wind shear exponent ranges are based on a 2021 study of lidar data from floating lease areas off the coast of New Jersey (Debnath et al., 2021). The wind veer ranges are based on a relationship with the shear exponent described in the same study (Debnath et al., 2021). The maximum for the turbulent wind standard deviation was selected based on a class B wind turbine as defined by IEC using the wind speed for each condition, and the minima were selected as having a turbulence intensity of 3.5% (IEC, 2019b). The range of air density values was taken as the largest seasonal variation for an offshore site (US East Coast) as identified in a 2019 study examining the effect of density changes on global power production (Ulazia et al., 2019).

The Justifications for the new floating/offshore specific parameter range selections are as follows:

- **Mooring mass:** The mooring mass range is taken as $\pm 1\%$ of the nominal value. This amount of uncertainty was recommended by industry experts due to manufacturing uncertainty and the addition of marine growth in a project lifetime.

- **System center of mass - upstream/upwind:** A range of $\pm 2\%$ was recommended by industry experts due to fabrication uncertainty. This value was applied to the nacelle, but could not be used for the platform which has a nominal value of 0.0 m. The platform range was chosen to result in a mass proportional shift to the shift in the nacelle. A large portion of the platform mass is ballast water, which is not shifted in the model adjustment, so the steel adjustment is increased to $\pm 8\%$ to account for the same change in the ballast position.

- **System center of mass - vertical, System inertia - pitch direction, System inertia - yaw direction, and System mass:** The same recommended range of $\pm 2\%$ was used for all the system mass and inertia properties. Again the platform steel has a larger shift of $\pm 8\%$ to account for the ballast water.

- **Water depth:** The range in water depth was based on changes in water level for a single location due to tides and storm surge. The limits of highest and lowest observed water levels come from the LIFES50+ study, which assessed the metocean conditions for three different types of floating wind sites (Gómez et al., 2015). Because these changes occur at a single site through time, no corresponding changes to the mooring system are made.

- **Water density:** The range in water density is a conservative range based on world limits of 1020.0 kg/m$^3$ to 1030.0 kg/m$^3$ from the NOAA World Ocean Atlas (NCEI, 2018).

- **Directional spreading:** The range in directional spreading was chosen based on expert opinion. The cosine spreading function with an equal-energy method was used as described in the 2014 paper recommending it for OpenFAST implementation (Duarte et al., 2014). The function is given in Equation 7, where the value of $s$ is set to 1.0, making a cosine squared function. The value of $C$ is set so that the total energy at each frequency is as desired for the spectrum (Duarte et al., 2014). This function $D(\Theta)$ is multiplied by the spectral value $S(\omega)$ for a combined frequency-direction magnitude.

$$D(\Theta) = C \left| \cos\left(\frac{\pi(\Theta - \Theta_{mean})}{2\Theta_{max}}\right) \right|^{2s} \tag{7}$$





– **Wind/wave misalignment:** The wind and wave misalignment range was selected to cover a conservative range of possible conditions. A 2014 study on wind and wave misalignment effects for floating wind loads found that the most tower fatigue loads came from direct alignment (Bachynski et al., 2014). A wide range of misalignment angles are considered to see the impact on other important loads.

– **Significant wave height:** The significant wave height is a function of wind and wave misalignment following joint probability distributions from a comprehensive metocean analysis of US offshore wind sites (Stewart et al., 2015). For each point in the input parameter hyperspace, a unique range of wave heights is applied based on the wind condition and the local misalignment. The minimum significant wave height is taken as 0.0 m for all wind conditions and misalignment angles (rather than what is shown in Figure 3). The maximum values are taken from the 2016 study based on data from

the National Data Buoy Center, and are the maxima of the three site for each misalignment angle (Stewart et al., 2015). These maxima are shown with the solid black lines in the top section of Figure 3. The nominal values are taken as the aggregate mean value shown with the dashed black lines. The colored lines in the figure, with three of each line type, mark the East Coast, West Coast, and Gulf Coast individually.

The significant wave height, peak period, and spectral shape factor are used to define the JONSWAP wave spectrum as

given by Equation 8 (ABS, 2016). The peak rotational frequency, $\omega_p$, is equal to $2\pi$ divided by the peak period.

$$
\begin{aligned}
S(\omega) &= \frac{5}{16}\frac{H_s^2\omega_p^4}{\omega^5}\exp\left[-\frac{5}{4}(\frac{\omega_p}{\omega})^4\right]\gamma^\alpha(1-0.287\ln\gamma)\\
\alpha &= \exp\left[-\frac{(\omega-\omega_p)^2}{2\sigma^2\omega_p^2}\right]\\
\sigma &= 0.07 \text{ when } \omega \le \omega_p\\
\sigma &= 0.09 \text{ when } \omega > \omega_p
\end{aligned}
\tag{8}
$$

– **Peak wave period:** The peak period ranges are a function of the significant wave height, and again are uniquely calculated for each point in the input parameter hyperspace. The maxima are chosen based on the 2016 US metocean project, and are shown as a function of wave height with the solid black lines in the bottom row of subfigures in Figure 3. The

minima are chosen based on the breaking wave limit and are shown as a function of significant wave height with the dashed-dotted black lines. Again the three colored lines of each line type are the values for the three comprehensive US locations based on joint probability distributions (Stewart et al., 2015). At large wave heights, more and more of the period data falls below the breaking limit, eventually even the maximum periods are lower than the limit. These breaking waves cannot be accurately modeled with mid-fidelity models like OpenFAST, but the selected ranges of significant

wave height do not fall into this region.



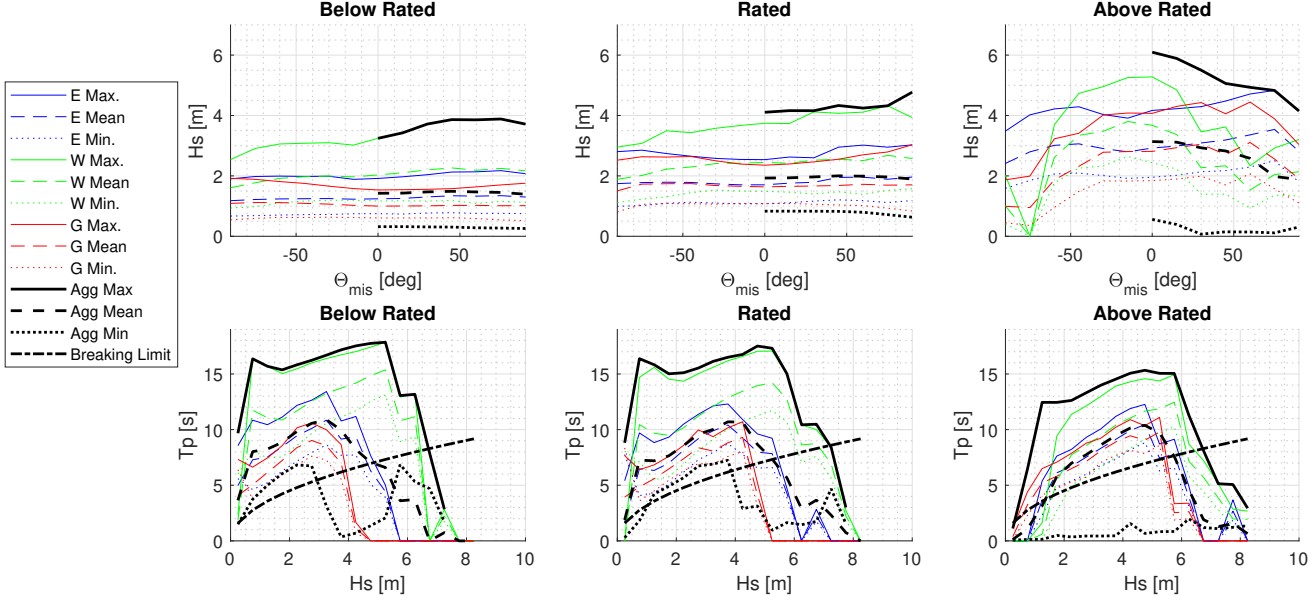

**Figure 3.** Wave height dependence on misalignment and wave period dependence on wave height from aggregate values (black) used from US East Coast (blue), West Coast (green), and Gulf Coast (red) comprehensive sites, data from (Stewart et al., 2015)

– **Wave spectral shape factor:** The wave spectral shape factor ranges were chosen following guidance from the American Bureau of Shipping, with a minimum of 1.0 and a maximum of 7.0 (ABS, 2016). The nominal values were selected as functions of the wave height and period as recommended by IEC and stated in Equation 9 (IEC, 2019a).

$$
\begin{aligned}
&\frac{T_P}{\sqrt{H_S}} \leq 3.6 \longrightarrow \gamma = 3.6 \\
&3.6 < \frac{T_P}{\sqrt{H_S}} < 5.0 \longrightarrow \gamma = 5.75 - 1.15\frac{T_P}{\sqrt{H_S}} \\
&5.0 \leq \frac{T_P}{\sqrt{H_S}} \longrightarrow \gamma = 1.0
\end{aligned}
\tag{9}
$$

– **Current speed:** The current speed range goes up to a maximum value for open ocean currents; this value of 2.0 m/s is from the Gulf Stream (Gyory et al.).

– **Current direction:** The current direction range goes up to a maximum of 90.0 degrees, perpendicular to the wind direction. This orthogonal current could create unique loading. It is expected that current coming from the downwind direction would have lower loads than when current loads are in the same direction as the wind.

– **Drag coefficient near water line:** In previous work analyzing this semi-submersible platform, the drag coefficient near the water line was identified as being particularly important (Wang et al., 2022). The water particle velocities are highest





here and higher order viscous effects are the most pronounced. The drag coefficient in this region was specifically isolated from the coefficients on the rest of the body to determine its unique influence. The selected nominal value was found to best replicate motions from model test experiments in previous work (Wang et al., 2022). The total range is based on the

possibly range of drag coefficients for cylinders as a function of Reynolds number. This coefficient is applied from the top of the columns down to an elevation 4.0 m below the mean water line.

- **Drag coefficient on column:** The nominal drag coefficient for the main length of the columns was again selected based on what was found to best replicate model test motions (Wang et al., 2022).

- **Drag coefficient on heave plate:** The nominal drag coefficient for the base of the columns, which act as heave plates

was also based on the previous work with this platform (Wang et al., 2022).

- **Axial drag coefficient:** The nominal axial drag coefficients for the columns was also based on the previous work with this platform (Wang et al., 2022). The range is larger as there is more variability with the unique viscous effects in the axial direction, compared to the well studied transverse drag on cylinders. Separated flow is immediate in this direction, and the drag forces are complex.

## 4 Output Quantities of Interest

Fourteen quantities of interest were selected to evaluate the importance of each input parameter. Table 4 lists these quantities of interest and their relevant labels and units. Outputs listed in red are most strongly linked to aerodynamics and the rotor, while quantities in black are more global.





**Table 4.** Output quantities of interest for load identification (red = rotor specific, black = global system)

| Quantity of Interest | Label | Units |
|---|---|---|
| Blade root bending moment | RootMp | N-m |
| Blade root pitching moment | RootMzc1 | N-m |
| Low speed shaft bending moment | LSSGagMp | N-m |
| Rotor torque | RotTorq | N-m |
| Yaw bearing bending moment | YawBrMp | N-m |
| Yaw bearing yawing moment | YawBrMzp | N-m |
| Tower base bending moment | TwrBsM | N-m |
| Blade tip out of plane deflection | OoPDefl1 | m |
| Generator power | GenPwr | W |
| Mooring line tension at fairlead | Fair | N |
| Mooring line tension at anchor | Anch | N |
| Watch circle | WatchCircle | m |
| Heel angle | Heel | degrees |
| Nacelle acceleration | NacAcc | m/s$^2$ |

The blade-root bending moment, yaw-bearing bending moment, tower-base bending moment, low-speed shaft bending mo-

ment, and watch circle each have components in two directions. For ultimate loads, the value of each quantity was taken as the maximum vector magnitude. For fatigue loads a load rose approach was used. The cycles were broken into twelve directional bins (15 degree increment), and the bin with the highest standard deviation was used for that outputs fatigue contribution.

The heel angle is a combination of the pitch and roll angle, and was combined at every time step for both ultimate and fatigue loads following Equation 10. The nacelle acceleration is the vector magnitude of the accelerations in all three directions for

both ultimate and fatigue loads.

$$Heel = \arctan\left(\sqrt{\tan(Pitch)^2 + \tan(Roll)^2}\right) \qquad (10)$$

For the mooring loads, the forces at all three fairleads and all three anchors were considered. The line with the largest maximum and the largest standard deviation were selected separately for the ultimate and fatigue loads on a case by case basis.





# 5  Results

In total, 324,000 OpenFAST and 72,000 TurbSim simulations were run following the quantities in Equation 11. Only 7 of the input parameter perturbations resulted in a change in the turbulent inflow wind calculations. For the other 28 perturbations, the inflow wind from the starting point was used reducing the number of required TurbSim simulations.

$$324,000 \text{ OpenFAST Runs} = (3 \text{ Wind Conditions})(30 \text{ Starting Points})(1 + 35 \text{ Input Perturbations})(100 \text{ Seeds})$$
$$72,000 \text{ TurbSim Runs} = (3 \text{ Wind Conditions})(30 \text{ Starting Points})(1 + 7 \text{ Wind Input Perturbations})(100 \text{ Seeds})$$
(11)

The histograms in Figures 4 and 5 show the ultimate and fatigue EE values, respectively, for all quantities of interest from all simulations. The plots are broken by color according to the wind speed condition. The red lines in each plot indicate the threshold for a significant event, determined by two times the standard deviation as described in Equations 5 and 6.

The ultimate values shown in Figure 4 are strongly stratified by wind condition for most outputs. The only outputs without this strong separation are the global system motion related quantities of mooring line tensions, watch circle, and heel angle. The fatigue EE values are much more consistent between wind conditions, which is interesting considering that each is weighted by the probability of the wind speed. The spread in the ultimate values is larger than for the fatigue values.

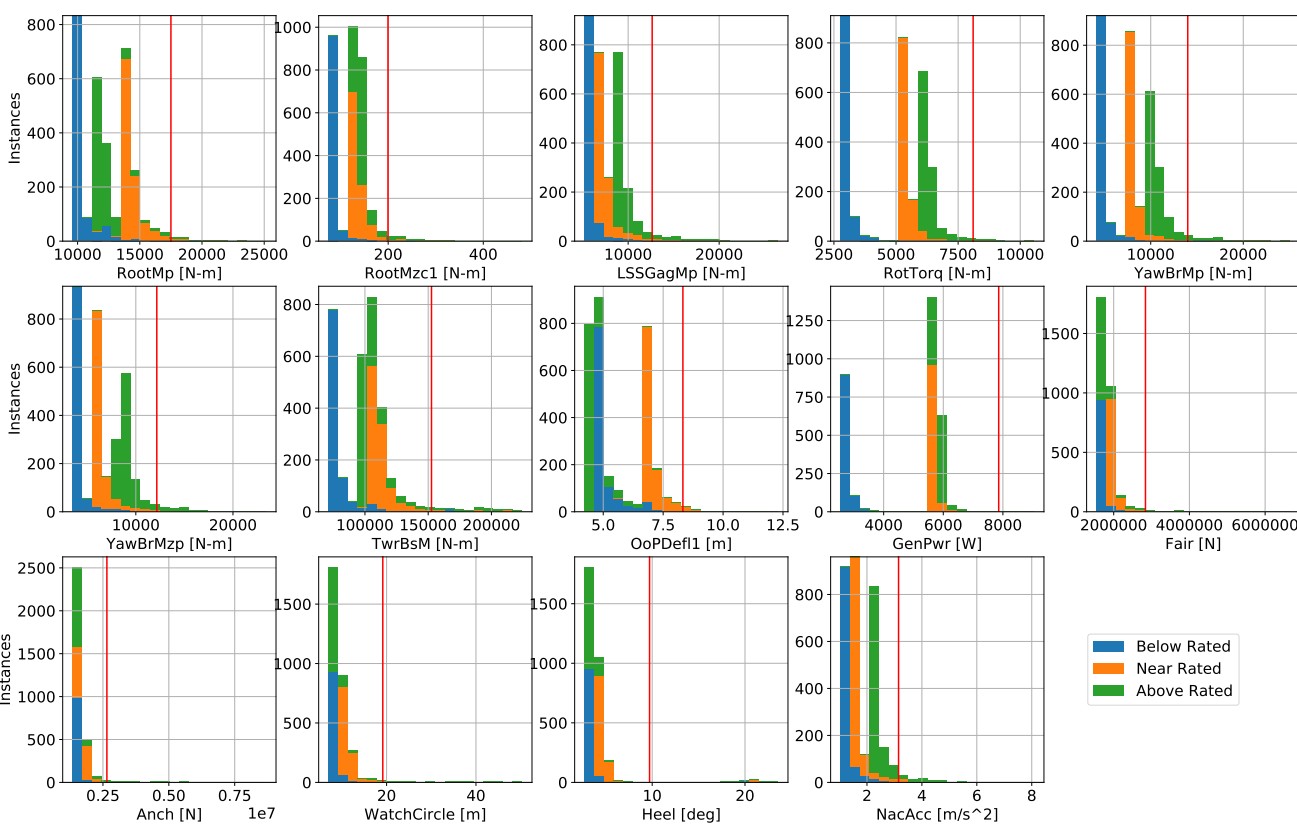

**Figure 4.** Ultimate EE value histograms broken by wind speed condition for 14 quantities of interest; red line marks threshold for significant EE value

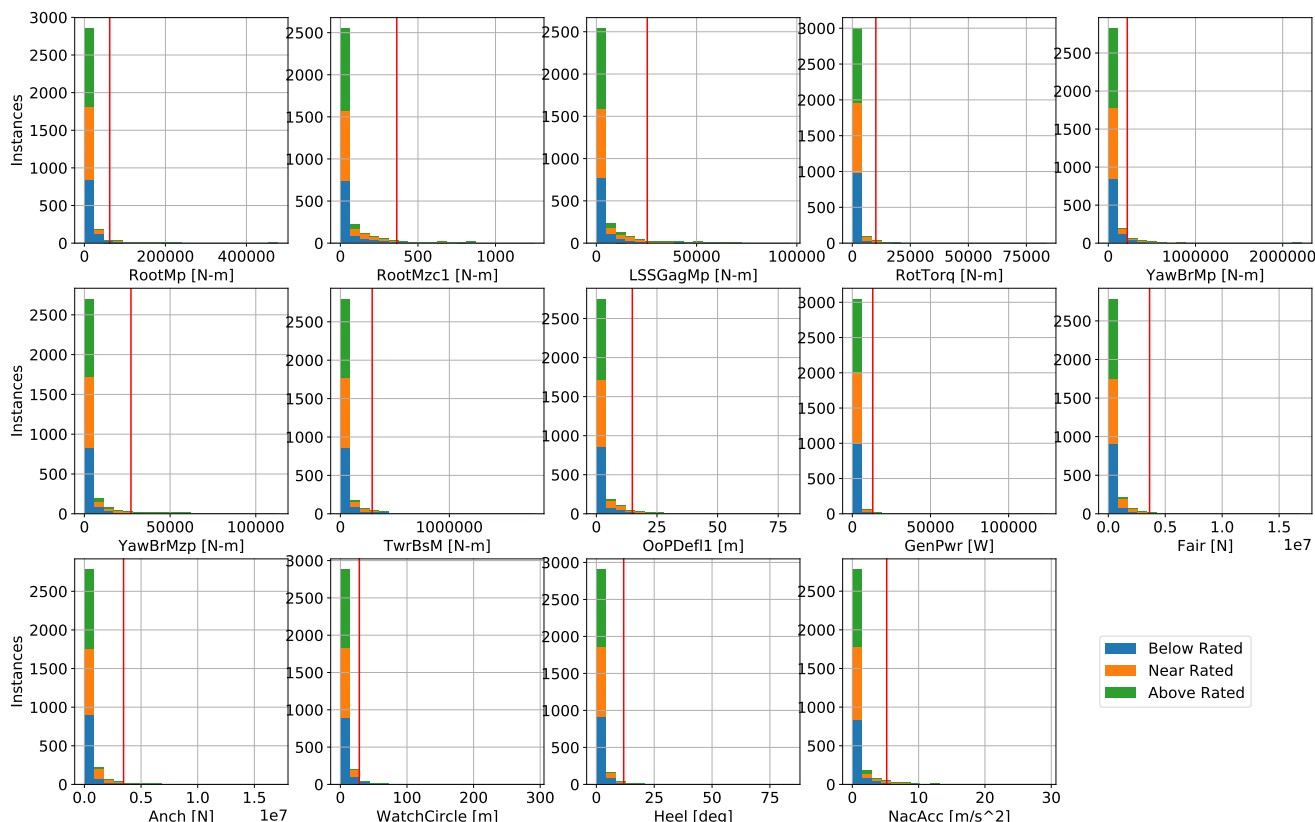

**Figure 5.** Fatigue EE value histograms broken by wind speed condition for 14 quantities of interest; red line marks threshold for significant EE value

The bar graphs in Figures 6 and 7 show the number of ultimate and fatigue significant events, respectively, attributed to each input parameter. The bars are divided and colored according to which output parameter the significant event is related to. The outputs closely linked with the rotor have no hatch, and the more global outputs have a diagonal hatch.

Looking at the ultimate load sensitivity in Figure 6, similar to previous land-based sensitivity studies, the wind speed standard
deviation in the main wind direction is the most influential input parameter. The next most important parameters are the horizontal system center of mass and the current velocity. The wind-related inputs (red) drove the rotor related outputs (no hatch) for the most part, and also have some important contributions to the nacelle acceleration. Most structural and mass properties have a limited impact on the ultimate loads, with the exception of the horizontal center of mass. This property is highly important for the extreme heel angle and the bending moment at the tower base. These two outputs are expected to be
coupled and are logically driven by this important input. The wave and current conditions appear to only drive the ultimate loads related to the platform translation. The watch circle displacement and the mooring tensions at the fairleads and anchors are




reasonably influenced by the current and drag coefficients, which contribute highly to mean forces. It is somewhat surprising how few significant events are attributed to the wave conditions. That said, it should be recalled that all wind conditions involved an operating wind turbine, which adds considerable damping. It is possible that the influence of the wave parameters
may be much stronger for idling load cases.

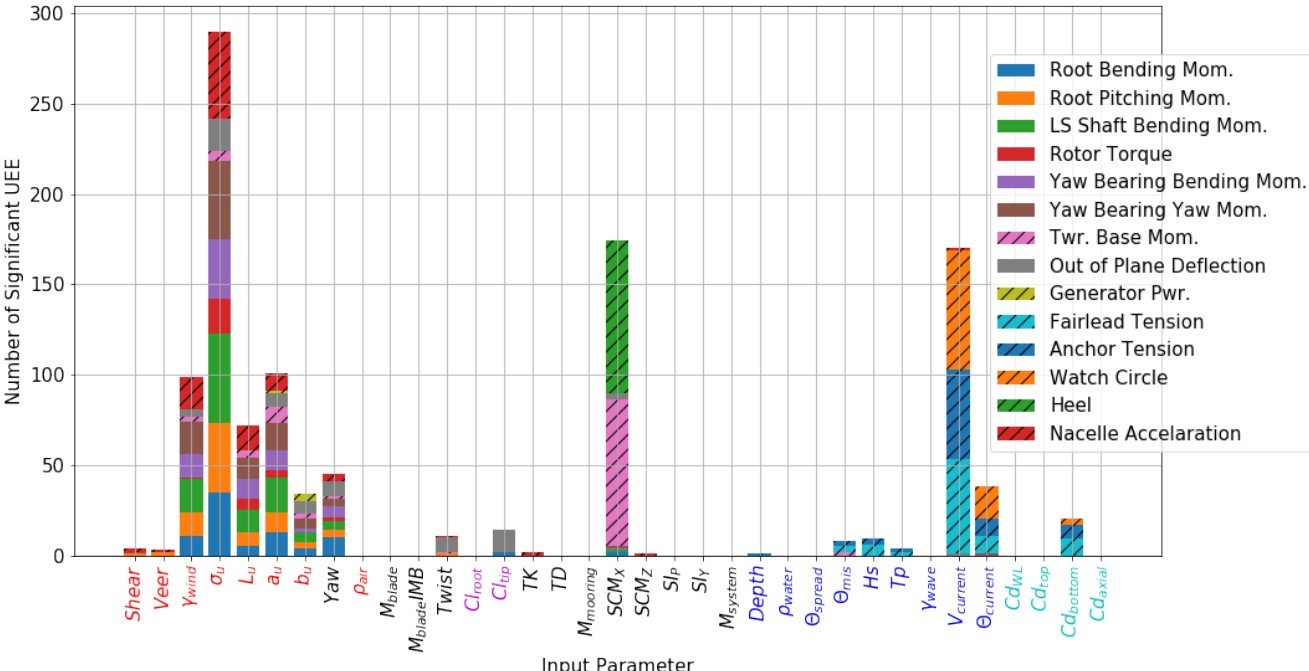

**Figure 6.** Floating offshore wind turbine ultimate load EE sensitivity, broken by load type, solid bars correspond to turbine specific responses and hatched bars correspond to global system responses (input parameter colors: red = inflow wind, magenta = aerodynamic forces, black = system and structure, blue = sea state, and cyan = hydrodynamic forces)

Looking at Figure 7, the standard deviation of the wind speed in the main direction is also the most influential input parameter for fatigue loads, again most strongly impacting the rotor-related loads. Compared to the ultimate load sensitivity, the wave parameters have a larger effect on the fatigue loads. The significant wave height, peak wave period, and wind and wave misalignment angle all are driving inputs for the global motion fatigue values of the platform. The wave conditions are more
important for the variability in the loads than the extreme values. The current speed and direction are also important for these fatigue values, which is less intuitive given their more constant forces. No structural or mass properties overly dominated the fatigue loads.

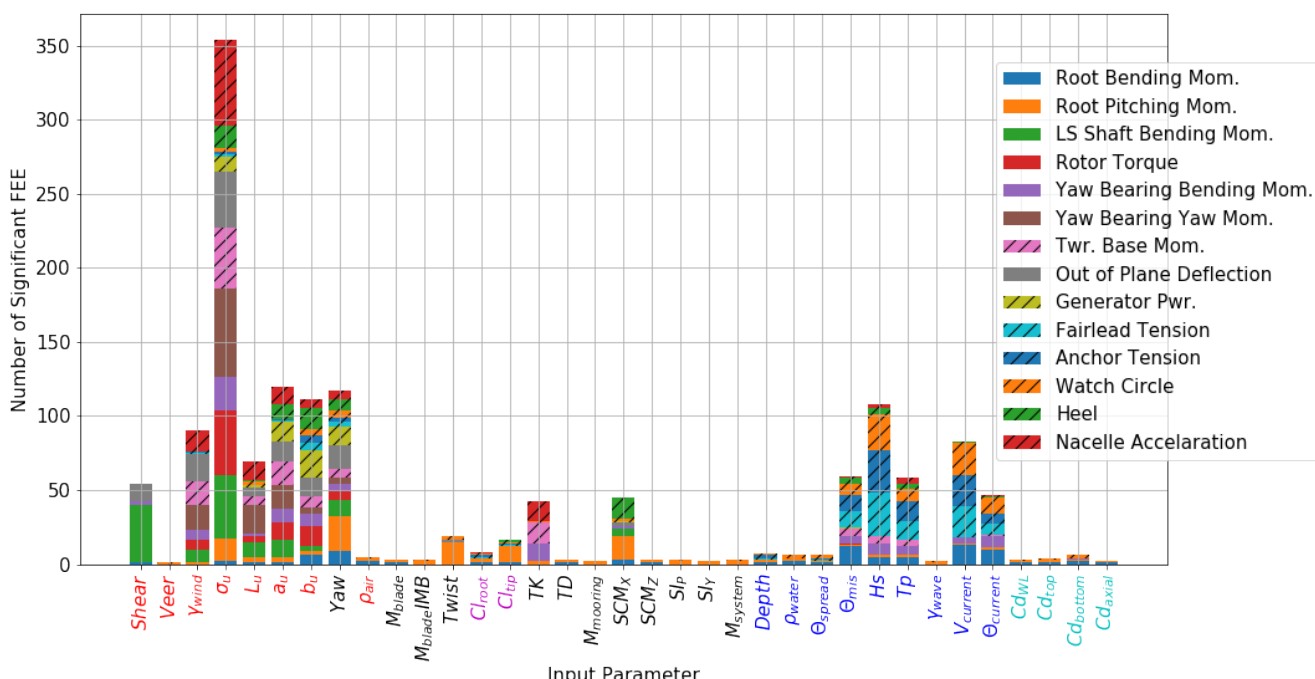

**Figure 7.** Floating offshore wind turbine fatigue load EE sensitivity, broken by load type, solid bars correspond to turbine specific responses and hatched bars correspond to global system responses (input parameter colors: red = inflow wind, magenta = aerodynamic forces, black = system and structure, blue = sea state, and cyan = hydrodynamic forces)

## 6 Seed Convergence

The number of seeds needed for stochastic irregular waves and turbulent wind environments is specific to a given platform,
turbine, and environmental condition. If the number of seeds is very low, it is likely that the differences in loads between
simulations may be more related to the difference in seed than to the perturbation in input variable. When the number of seeds
is high enough, the averaging process in Equations 1 and 2 removes the dependence on the seeds, and the difference in output
can be accredited to the difference in input. Some inputs and some outputs are more strongly dependent on the number of
seeds, so the required number of seeds is not universal; however, the same number of seeds was used for each perturbation in
this study.

Given the very large number of inputs, outputs, starting points, and wind speeds, it is difficult to look at the influence of
the seed number for each input and output combination individually. Some examples of seed convergence for input and output
combinations that led to a large number of significant sensitivity are shown in Figures B1, B2, B3, and B4 in Appendix B.
The plots show the ultimate and fatigue output according to Equations 1 and 2, as a function of the number seeds run. The
blue line shows the results for the starting point with no perturbation, and the red line shows the results for the relevant input
perturbation. The number of seeds is sufficient when the difference between the blue and red lines is clearly larger than the



variability in either line due to seed. This means that the EE value is actually due to the input perturbation and not the chosen seeds.

Ultimately, the important criteria is that the identified most influential input parameters are not dependent upon the seeds
used. This means that the bar plots in Figures 6 and 7 should not change if more seeds are added or if different seeds are used. Figures 8 and 9 show this convergence of the final relative sensitivities, measured by number of significant events for ultimate and fatigue loads, respectively. 100 seeds were run for each point and perturbation in the hyperspace. The post-processing calculations were performed pulling from a pool of 1 to 100 seeds, and the resulting number of significant ultimate and fatigue events are plotted for each input. The final point in the lines matches the bar heights in Figures 6 and 7. When doing this
convergence check, the pools of $S$ seeds are taken randomly from the full set of 100, so that the results at $S+1$ are not as strongly tied to the results at $S$, leading to a smoother, more clear convergence.

The ultimate loads sensitivity for some important input parameters is almost independent of the number of seeds, including $SCM_X, V_{current}, \gamma_{wind}, a_u,$ and $L_u$. The relative importance of $Shear$ and $Veer$, in particular, is much higher when only few seeds are used. The ultimate loads sensitivity converges quickly for all inputs except $\sigma_u$, with stable relative levels after about
40 seeds. After this point, the variations are small compared to the differences between inputs. However, $\sigma_u$, the parameter leading to the highest sensitivity, continues to become relatively more important with an increased number of seeds. Between 80 and 90 seeds it is evident that the number of significant events is converging towards just below 300. Given the large difference from the other input parameters, this provides sufficient insight. The number of seeds needed to identify sensitivity, especially for $\sigma_u$, is much higher than the minimum of 6 recommended by IEC (IEC, 2019a). Wind turbulence has high energy at very
low frequencies, resulting in some large amplitude, low frequency cycles in any given time series, and strongly increasing the dependence on seed number.





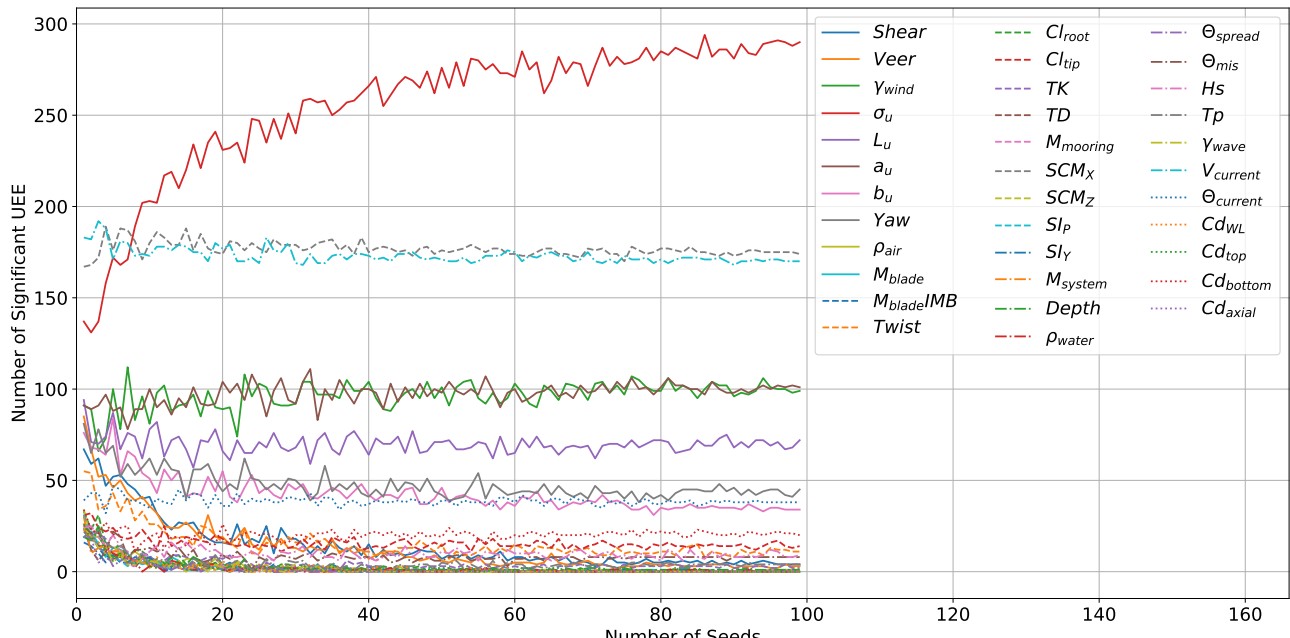

**Figure 8.** Convergence of ultimate response sensitivity due to number of random seeds

The fatigue loads sensitivity results converge much quicker than the ultimate loads sensitivity results. Even for $\sigma_u$, the number of significant events is relatively stable after only 20 seeds. It is expected that the fatigue would require fewer seeds than ultimate loads, as the cyclic amplitudes over a time series are less likely to have outliers than the tails of the statistical distribution.






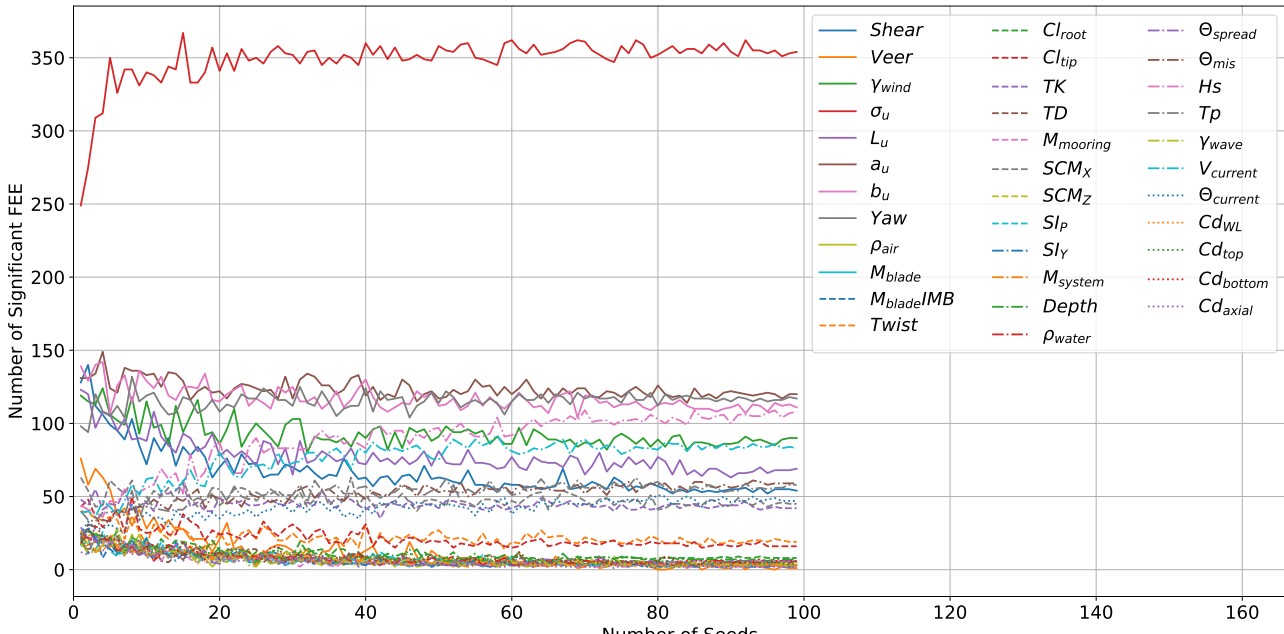

**Figure 9.** Convergence of fatigue response sensitivity due to number of random seeds

The sensitivity results are determined to be converged after 90 seeds for ultimate loads and after 30 seeds for the fatigue loads. This process is likely load case and device dependent, so it should be performed independently for each application of the EE sensitivity analysis approach. The results shown in Figures 4, 5, 8, and 7 use the data from all 100 run seeds.

## 7 Starting Point Convergence

The radial one-at-a-time perturbation method looks at the uncoupled sensitivity at a single point in the parameter hyperspace, within the ranges defined. Each point, chosen following the Sobol sequence, likely has a different local sensitivity value. To get a picture of the true sensitivity across the full domain, a sufficient number of starting points needs to be used. It should be noted that while the process treats the inputs as fully uncoupled (with the exception of wave misalignment, height, and period), there are likely some combinations of inputs that would not be physically expected, and the sensitivities at these points can still influence the findings.

The number of necessary starting points is a function of the local second partial derivatives. When the derivative changes sharply through the domain, more starting points are needed for converged identification of the most important inputs. Similar to the method of determining seed convergence, the sufficient number of starting points was determined by calculating the number of significant events per input using a range of 1 to $B$ starting points. Figures 10 and 11 show the convergence of the significant events with respect to the number of starting points. The number of EE values is a direct function of the number of





starting points, so as the number of starting points grows, so does the number of significant events. The fraction of the total number of significant events is plotted instead of the absolute number to be able to compare and track convergence.

In this case, the sets of $B$ starting points are not random, but follow the Sobol sequence which is designed to fill in the n-dimensional region with an even distribution. For the seed convergence, each individual seed had no inherent bias towards

one input parameter. However, each starting point does have a bias toward the local partial derivative, resulting in a less smooth convergence path. Still, if enough starting points are used, the global sensitivity will converge so that additional points do not effect the conclusions.

For the ultimate load convergence shown in Figure 10, after 20 starting points it is clear that $\sigma_u$ is the most dominant input. $V_{current}$ and $SCM_X$ clearly have secondary importance, and the rest of the inputs are far less influential. It appears that the

relative impact of $\sigma_u$ may still grow some if more than 30 starting points are used, but the order of importance seems unlikely to change.

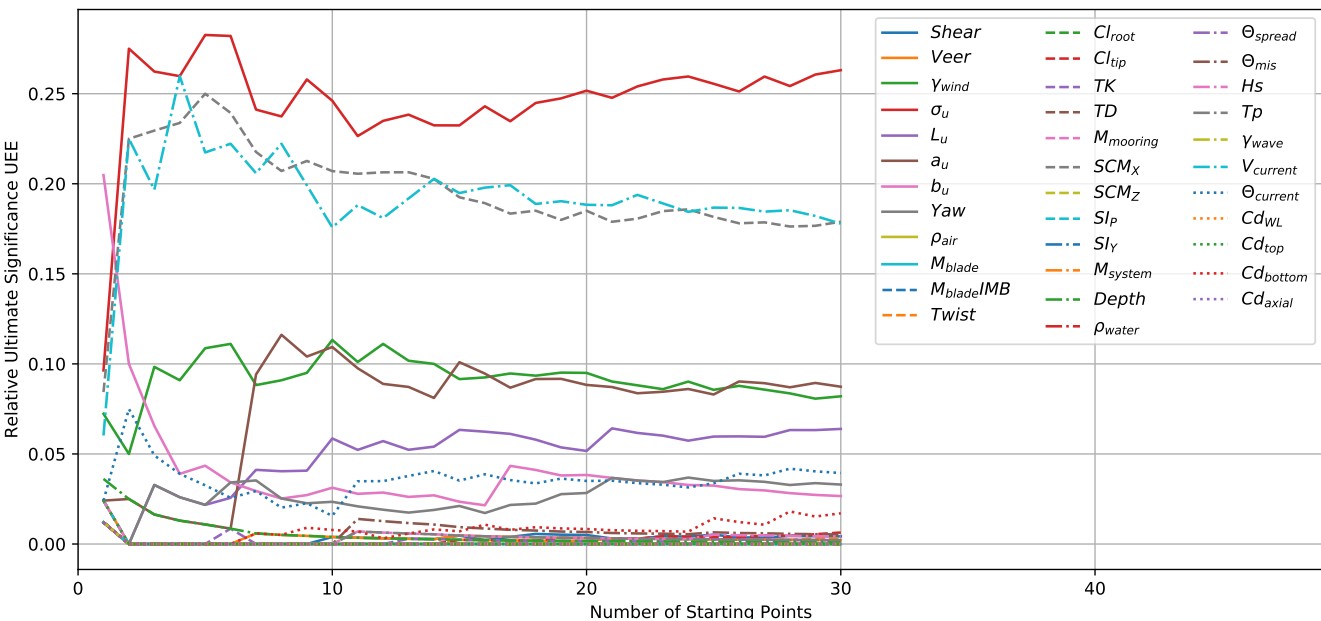

**Figure 10.** Convergence of ultimate response sensitivity due to number of input parameter hyperspace starting points

For the fatigue load convergence shown in Figure 11, $\sigma_u$ is even more clearly dominant, even after only 10 starting points. At 30 starting points, it appears there is no relationship between the relative sensitivity and additional starting points.



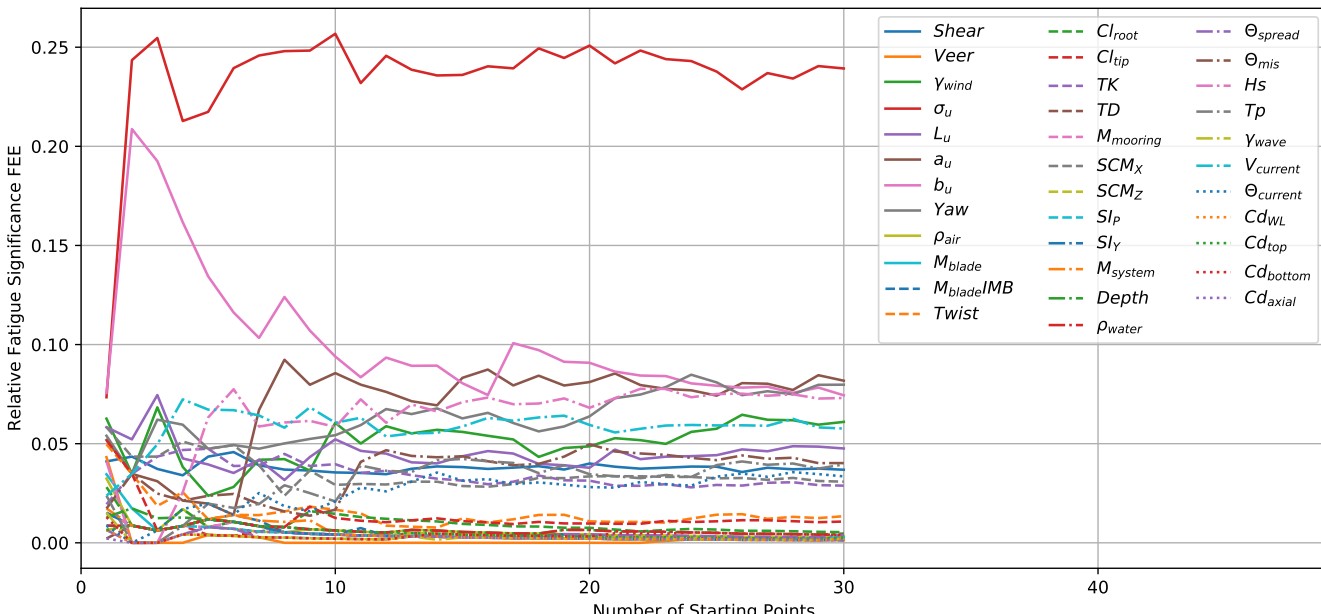

**Figure 11.** Convergence of fatigue response sensitivity due to number of input parameter hyperspace starting points

## 8 Conclusions

This work used a radial EE approach to identify which numerical modeling input variables have the most important effects on ultimate and fatigue loading of the DeepCWind floating platform supporting the NREL 5-MW offshore wind turbine. The standard deviation of the load was used as a proxy for the fatigue. All modeling parameters have a range of validity, and this process can determine which uncertainty ranges should be assessed in greater detail. A total of 35 input parameters were tested and evaluated using 14 output responses. The results were delineated by output to understand which specific input and response

relationships have the highest sensitivity. The required number of seeds used for stochastic irregular waves and turbulent wind environments was assessed to ensure that the variability due to seed was not influencing the conclusions. The required number of starting points in the parameter range domain was also assessed to ensure that the sensitivity assessment approximated a global sensitivity. In total, 324,000 OpenFAST simulations and 72,000 TurbSim simulations were run, spanning three different operating wind speed conditions.

The evaluated input parameters included wind and water environment descriptions, structural properties, and aerodynamic and hydrodynamic modeling coefficients. All parameter ranges were assessed to cover the possible variation either due to changes in time, uncertainty in construction, or uncertainty in precision.

It was found that ultimate loads EE values were highly stratified by wind speed; depending on output load, either the above-rated or near-rated condition contributed the most extreme load sensitivity. Significant fatigue EE values, however, were not

clearly split by wind condition.





The EE approach has been shown to be effective for screening the most influential modeling parameters for FOWT load assessment. For the combination of the NREL 5-MW offshore wind turbine on the DeepCWind semi-submersible, the input parameters contributing to the highest sensitivity in **ultimate** loads are:

– Primary: $\sigma_u$

– Secondary: $SCM_X$ and $V_{current}$

– Tertiary: $\gamma_{wind}$, $L_u$, $a_u$, $b_u$, $Yaw$, and $\Theta_{current}$

and the input variables contributing to the highest sensitivity in **fatigue** loads are:

– Primary: $\sigma_u$

– Secondary: $\gamma_{wind}$, $a_u$, $b_u$, $Yaw$, $Hs$, and $V_{current}$

– Tertiary: $Sheer$, $L_u$, $TK$, $SCM_X$, $\Theta_{mis}$, $Tp$, and $\Theta_{current}$

Similar to previous analyses with land-based wind turbines, the turbulent wind speed standard deviation in the main direction ($\sigma_u$) is the input parameter with the highest impact. Not only are rotor-specific loads very sensitive to this value, but so are the global platform motions. While $\sigma_u$ is the most important, all wind turbulence parameters have a significant impact on ultimate and fatigue loads.

Mooring loads and device footprint are dominated by current. Almost no significant events in these outputs come from the wind or wave variables. When it comes to mooring design, a significant effort should be placed on assessing the true range of current speeds and directions.

With the exception of the horizontal center of mass, the system mass, inertia, and structural properties have a lower impact on the loads; environmental conditions variability seems to be much more important. This is likely a platform-specific conclusion; 425 the DeepCWind semi-submersible is a relatively stable design with a large water-plane area and resonant frequencies outside of the main wave and wind energy. The one structure parameter that does have a high relative sensitivity is the horizontal system center of mass in the wind direction. This value is less important for the fatigue cyclic amplitudes, but is by far the main driver for the extreme platform heel angle, and subsequently, the tower-base bending moment.

While the wave misalignment angle, height, and period do have meaningful influence on some fatigue values, it is somewhat 430 surprising how little of an impact the wave conditions have on the extreme ultimate loads. The wave input perturbations contribute far fewer significant events than the current inputs, and even fewer compared to the turbulent wind characteristics. This would indicate that significantly more emphasis should be placed on understanding the wind environment than the wave environment for an offshore wind site assessment for this floater. Future analyses that includes idling conditions past the cut-out wind speed may find that wave inputs are more influential. It is likely that without the aerodynamic damping of the operational 435 turbine, changes in the wave height and period may lead to more considerable changes in the ultimate loads on the platform. It is still logical that the waves are more influential for fatigue loading than ultimate loading. Potential wave over-topping and





slamming events could also be design drivers for platform stiffener design, however these loads are difficult to understand using mid-fidelity modeling tools.

This method used in this project can be used as an important step in the design process. The results identify which input parameter uncertainty ranges need to be given particular attention. The conclusions should be treated as unique to the individual platform and turbine and it is recommended that non-operational load cases are also considered.

## Appendix A: Fatigue EE Value Histogram

Figure A1 shows the same histograms of fatigue EE values as shown in Figure 5, but with a zoomed in y-axis to see the upper end of the distributions. The lowest bin had by far the most instances for all quantities of interest, including contributions from all three wind speed conditions. The spread in the fatigue EE values is small compared to the ultimate EE values, and there is little stratification; this is clear in the full figure, but the distribution and significant events that surpass the threshold are not visible. The zoomed version of the figure shows this. Note that the lowest bin is made of instances from all three wind speed conditions, but the contributions from the near-rated and above-rated conditions are cropped out in this view.



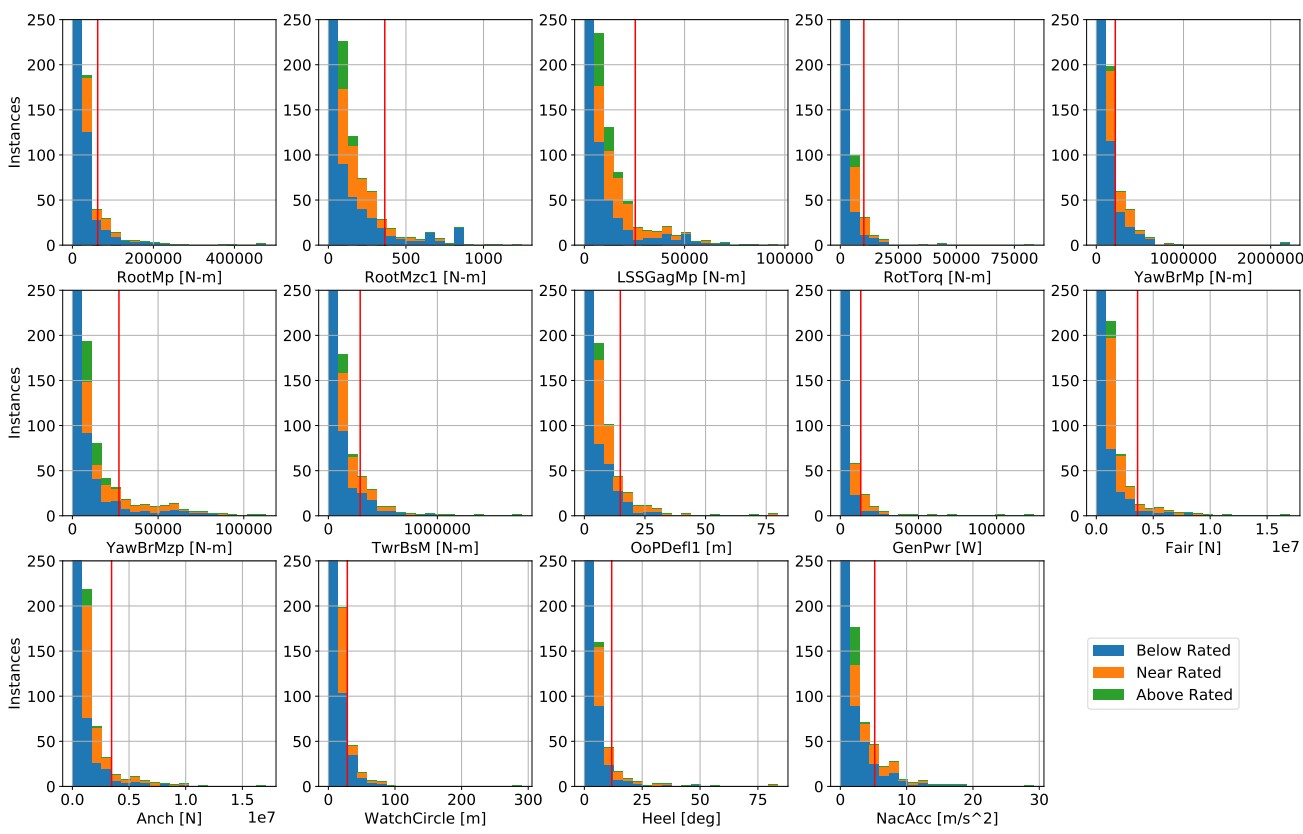

**Figure A1.** Fatigue EE value histograms broken by wind speed condition for 14 quantities of interest with a **zoomed y-axis**; red line marks threshold for significant EE value

## Appendix B: Seed Convergence

Figure B1 shows the seed convergence for the specific relationship of heel angle sensitivity to the system horizontal center of mass. The difference between the red line and the blue line indicates the local partial derivative, and the variability in each line indicates the effect of the seed. Convergence is achieved when the variability due to seed has disappeared, but a lower and more important threshold is when the difference between the lines is clearly distinguishable from the variability due to seed. Note that when the sensitivity between an input and an output is low, the difference due to the input perturbation may never be large compared to the fluctuations from the seed. This is true for the bottom-middle and bottom-right plots, that is, for the fatigue load in the near-rated and above-rated conditions. While $SCM_X$ is very important for the extreme absolute value of the heel, it has a much smaller impact on the cycle size. In general this relationship is relatively robust with respect to the number of seeds.



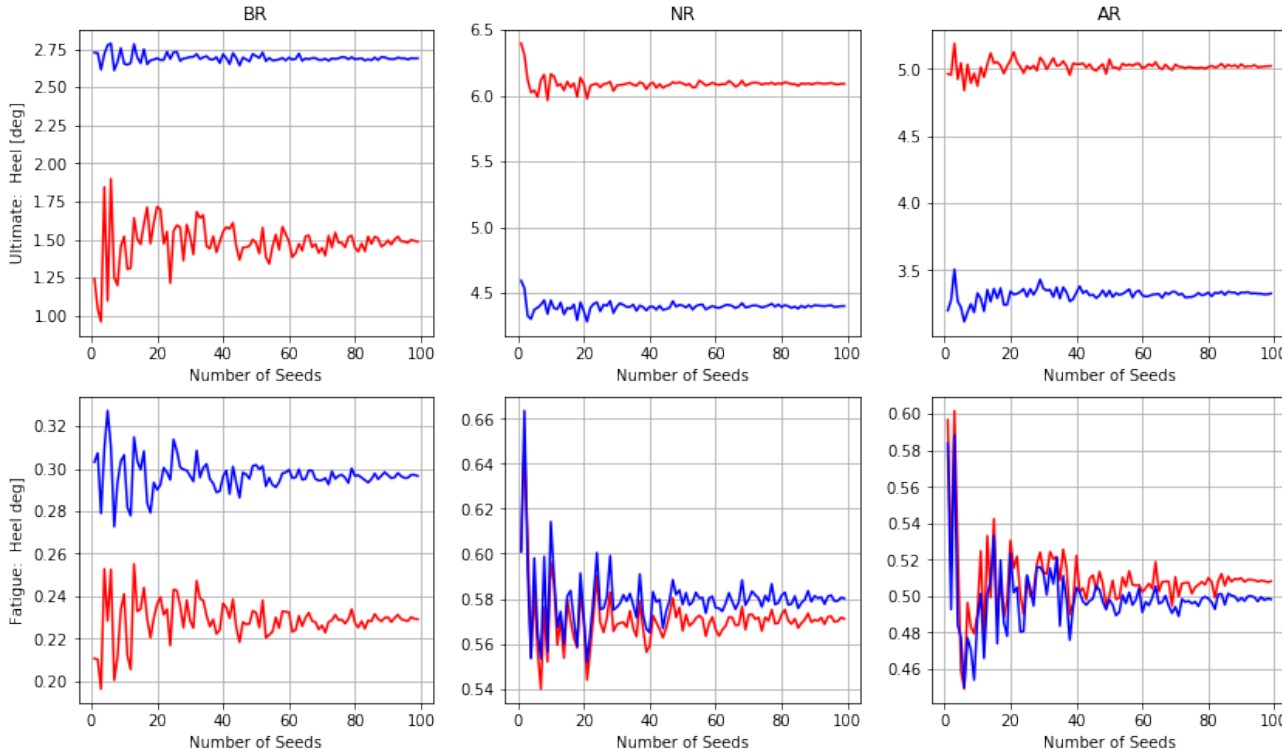

**Figure B1.** Seed convergence of platform heel due to system horizontal center of mass, Blue line: nominal starting point, Red line: perturbation in $SCM_X$

Figure B2 shows the seed convergence for the specific relationship of low speed shaft bending moment sensitivity to the turbulent wind speed standard deviation in the main wind direction. This relationship is much more sensitive to the number of seeds. The variability due to seed is large compared to the difference due to input perturbation. This is why the trend for $\sigma_u$ in Figure 8 is the slowest to converge. Note that this input-output relationship contributes a large number of significant events both for ultimate and fatigue loads. Even though the relationship is sensitive to the number of seeds, the difference between the red and blue lines is clearly distinguishable from the individual fluctuations by around 70 seeds.



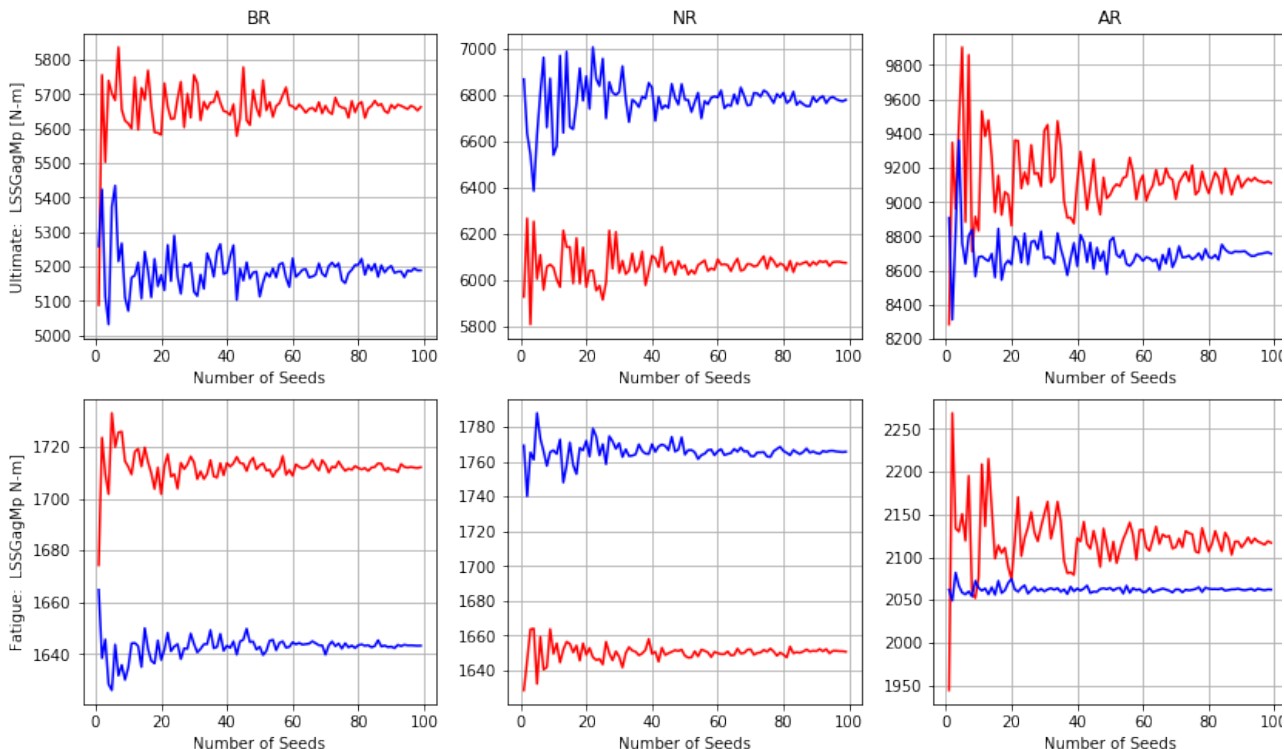

**Figure B2.** Seed convergence of low-speed shaft bending moment due to turbulent wind speed standard deviation in the main wind direction, Blue line: nominal starting point, Red line: perturbation in $\sigma_u$

Figures B3 and B4 show the seed convergence for two more important input-output relationships. Both of these relationships contribute to the high sensitivity to $\sigma_u$, and demonstrate that the influence of the perturbation is clearly distinguishable from the influence of the seed number when at least 80 seeds are used.

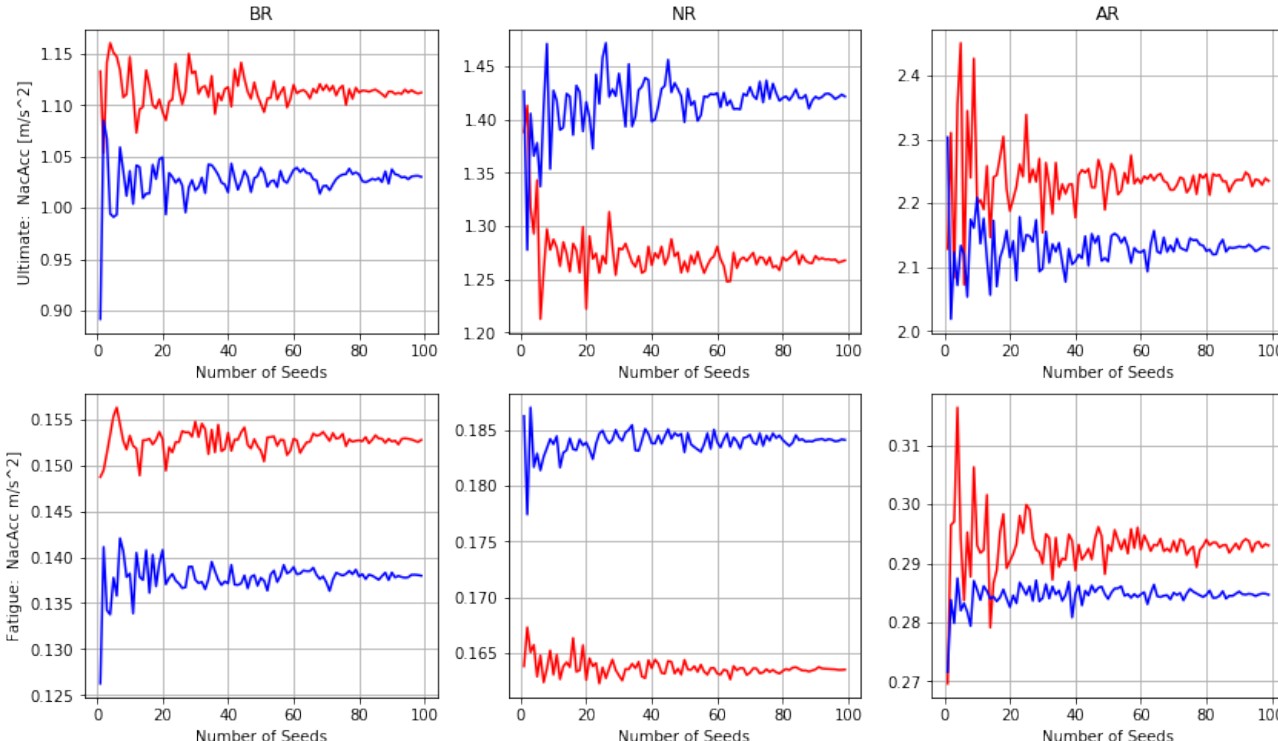

**Figure B3.** Seed convergence of nacelle acceleration due to turbulent wind speed standard deviation in the main wind direction, Blue line: nominal starting point, Red line: perturbation in $\sigma_u$

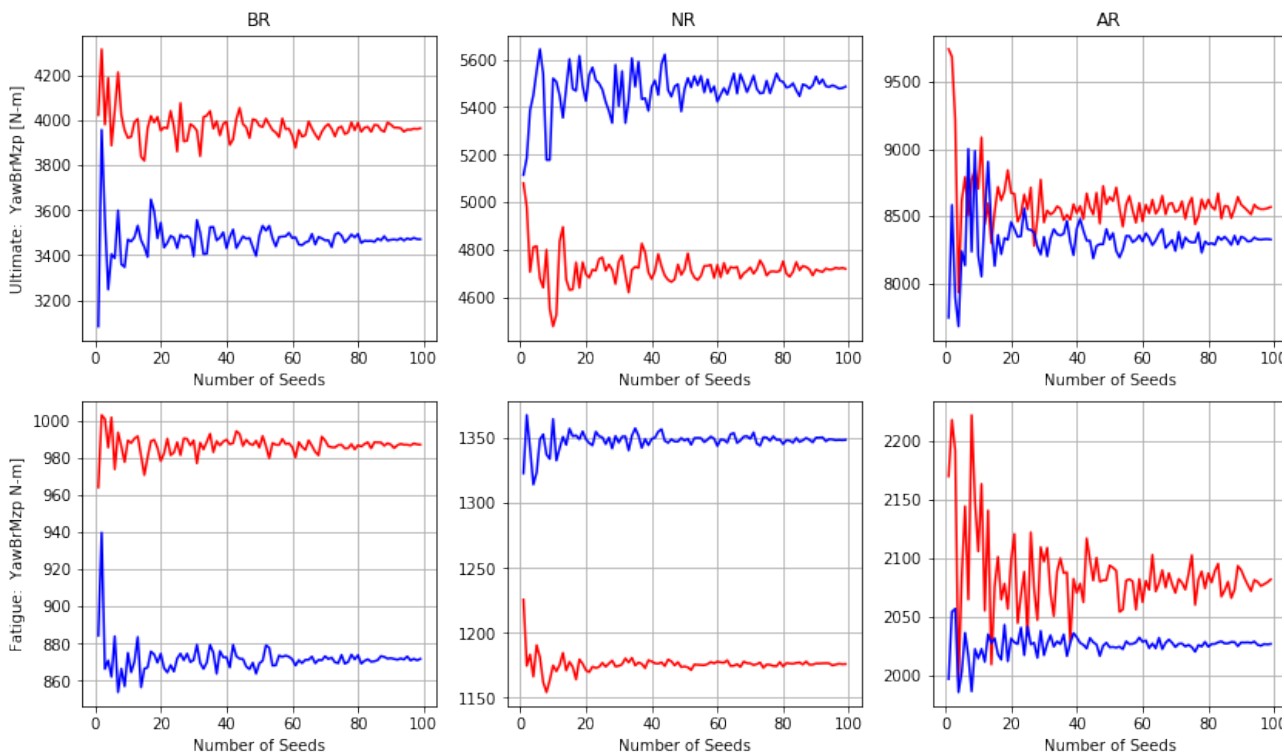

**Figure B4.** Seed convergence of yaw-bearing yawing moment due to turbulent wind speed standard deviation in the main wind direction, Blue line: nominal starting point, Red line: perturbation in $\sigma_u$

*Competing interests.* The contact author has declared that none of the authors has any competing interests.

*Acknowledgements.* We would like to thank Lu Wang of the National Renewable Energy Laboratory for performing the potential flow
calculations used in for the OC4-DeepCWind platform used in this analysis.

A portion of the research was performed using computational resources sponsored by the Department of Energy's Office of Energy Efficiency and Renewable Energy and located at the National Renewable Energy Laboratory.

This work was authored by the National Renewable Energy Laboratory, operated by Alliance for Sustainable Energy, LLC, for the U.S. Department of Energy (DOE) under Contract No. DE-AC36-08GO28308. Funding provided by the DOE Water Power Technologies Office.
The views expressed in the article do not necessarily represent the views of the DOE or the U.S. Government. The U.S. Government retains and the publisher, by accepting the article for publication, acknowledges that the U.S. Government retains a nonexclusive, paid-up, irrevocable, worldwide license to publish or reproduce the published form of this work, or allow others to do so, for U.S. Government purposes.



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
