# Peer review of "Sensitivity Analysis of Numerical Modeling Input Parameters on Floating Offshore Wind Turbine Loads"

_Wind Energy Science, 2023_

## Author Comment (AC1)

Dear referees,

Thank you very much for your time providing thorough feedback and suggestions for our paper. Please find our answers to your comments and the corresponding updates to the paper below. We hope to have addressed all of your comments. This has improved the manuscript. We will be happy to have continued discussion. We will submit the revised manuscript including a pdf highlighting the differences made to the text in the next few days.

Thank you,

Will, Jason, and Amy

**Referee 1**

Major Comments:

*First and foremost, I have significant concerns about the treatment of fatigue loads. As you write in the Conclusion (but not the abstract), "The standard deviation of the load was used as a proxy for the fatigue." I understand that you did not want to complicate the post-processing or start adding material uncertainties to your input parameters, but to actually draw conclusions on fatigue loads, you need to calculate fatigue loads. The aggregation of fatigue loads is very nonlinear and not treating this rigorously is likely impact your results.*

*If you do not want to post-process your results into fatigue loads, then I would recommend removing references to "fatigue loads" and instead referring to "load variability". But in either case, you should revisit Eq. 4. If you have fatigue loads, you should not be aggregating them with a mean operator but rather with a damage-equivalent-load-like operator. If you keep the standard deviation, then that should be aggregated by the standard deviation, not by the mean. Either way, the fatigue sensitivity calculation in Eq. 4 should be revisited. I don't expect this would impact the ranking, but it might impact the relative importance of certain parameters.*

We agree that this needs to be more clear throughout the paper instead of only discussed in the Approach section. As you mentioned, we didn't want to start introducing uncertainties in material properties, but instead focus on the loads. Our assumption is that the frequencies and amplitude distributions for a given load will be similar for each parameter perturbation (given that frequencies are heavily impacted by rotor harmonics), and that the standard deviation can capture the magnitude of those amplitudes. Emphasis was added to the conclusion that perturbations with a large influence on frequency may have an under-identified sensitivity with this approach. It is also added that no non-linear fatigue paths are considered. The term 'fatigue-proxy' is used in all instances in the document to make sure it is known that the comparison is not for a true fatigue value, without reading the full text.

The aggregate in Equation 2 takes the mean only across the seed numbers for a specific perturbation. Looking at the standard deviation across the seed numbers would give a measurement of the variability due to seed number, but would lose the variability due to the perturbation. In Equation 4 a relative comparison is used scaling the fatigue-proxy values by the probability of occurrence as would be done in a typical fatigue calculation.

All mentions of 'fatigue' have been edited to say 'fatigue-proxy' throughout the paper.

In the first paragraph of the conclusion - "The standard deviation of the load was used as a proxy for the fatigue. This simplification quantifies load variability but likely under-predicts the sensitivity to input parameters which strongly influence load frequency. Non-linear fatigue paths are not included in this method at all."

*Second, your paper does not reference existing literature that considers loads sensitivities of floating wind turbines. This seems like a rather unfortunate oversight, as the conclusions of the existing literature could be compared with the conclusions that are drawn in your paper. Here are three references I found that seem relevant, but there may be more:*

*Doubrawa, Paula, et al. "Load response of a floating wind turbine to turbulent atmospheric flow." Applied Energy 242 (2019): 1588-1599.*

This 2019 paper compares two commonly used wind turbulence models. Comparisons are made to high fidelity large eddy simulation results. Our paper only uses a single wind turbulence model, and varies the parameters of it, as well as many other non-wind related input parameters. The results from this 2019 paper are interesting and highlight the unique sensitivity of floating platforms to wind turbulence modeling, but it is difficult to compare results with our work.

*Nybø, Astrid, Finn Gunnar Nielsen, and Marte Godvik. "Sensitivity of the dynamic response of a multimegawatt floating wind turbine to the choice of turbulence model." Wind Energy 25.6 (2022): 1013-1029.*

This 2022 paper is similar to the previously commented Doubrawa et al. paper; it focuses specifically on two commonly used turbulence models for mid-fidelity analysis (Kaimal, Mann), and compares the results to higher-fidelity tools (LES and TIMESR). The findings again highlight that floating platforms have an especially high sensitivity to turbulence model, and that mid-fidelity models are potentially lacking. Again the focus of this work is different enough that it is difficult to make comparisons.

*Müller, Kolja, and Po Wen Cheng. "Application of a Monte Carlo procedure for probabilistic fatigue design of floating offshore wind turbines." Wind Energy Science 3.1 (2018): 149-162.*

This 2018 paper only varies three input parameters: wave height, wave period, and wind speed. The work provides interesting information of how many simulations are required for converged knowledge of loads, however it is difficult to compare to our paper. In addition to the small number of variable input parameters, the desired knowledge from this 2018 paper is the uncertainty in fatigue load, but not necessarily the sensitivity to a given parameter.

*Third and final, I'm not sure the paper adequately explains the connection between the sensitivity of certain input parameters and the shift in UEE caused by the shift in mean. for example, in Line 297 you write*

*"The ultimate values shown in Figure 4 are strongly stratified by wind condition for most outputs. The only outputs without this strong separation are the global system motion related quantities of mooring line tensions, watch circle, and heel angle"*

*This to me seems logical, as many of the load channels have a substantially different mean value at different wind speeds. But the text does not connect this to $Y_w$ in Eq. 3, which is of course the value that*

*can change a lot. I expect also that this is why the anchor and fairlead ultimate tension are more sensitive to the current than to the wave conditions, because the current shifts the mean load.*

We want to note that the value added in Eq. 3, Yw_bar, is the output based on the nominal value for all input parameters, not the mean output value. We agree that the mentioned stratification is largely due to the shift in the nominal output, but we still believe that this is a logical way to assess the sensitivity for the ultimate load. A high ultimate load sensitivity for an ultimate load that is relatively low compared to other wind conditions is not a concern. More discussion of this influence has been added to Section 5.

Third paragraph of Section 5 - "This separation is largely influenced by the changes in the nominal output, $\overline{Y_w}$, added to the EE value as shown in Equation 3. The ultimate EE value for the blade loads of the root bending moment and tip deflection are highest for the near-rated condition, logically where thrust is expected to be highest. The tower, shaft, and nacelle ultimate EE values are highest for the above-rated condition. The only outputs without a strong wind speed stratification are the global system motion-related quantities of mooring line tensions, watch circle, and heel angle. These three outputs are later shown to be largely dominated by current velocity and system center of mass, both input parameters that are independent of wind speed."

*I have a few other comments that I will mention here purely as food-for-thought for your future work. First, although the NREL 5 MW is, as you note, a well-characterized model, it is becoming an outdated design, so you might consider upgrading to the IEA 15 MW UMaine semisub.*

The authors agree and expect that the results could change with a larger and lighter system. Future work is planned to look at larger turbines and floaters. The advantage to this platform is the large amount of previous analysis and documentation including previous sensitivity studies. This allows for comparison of parameter importance between land-based and offshore. The demonstrated approach is more important than the specific results, and this design is very well known so readers can understand the context. It is expected that each design and conditions investigated could have different sensitivities. The OpenFAST model for this system is also very robust, avoiding physical or numerical instabilities across a wide range of parameters.

*Second, you are using ElastoDyn, which is of course a linear structural model. It could be interesting to see the sensitivity of the loads to modelling with BeamDyn, to see if there are any significant changes. But these changes might be insignificant in a stiff blade design like the NREL 5 MW.*

Addition to the first paragraph of Section 2.2 - "Previous analysis of the NREL 5-MW baseline wind turbine has shown that the higher-fidelity BeamDyn solver produces results that are generally aligned with the lower-fidelity ElastoDyn solver. The largest difference for this turbine is that BeamDyn does predict a small amount of blade torsion not captured by ElastoDyn, but this is generally compensated by the active blade-pitch controller."

Minor Comments:

- *Line 69: It could be nice to include a description of the upcoming sections, especially mentioning Sections 6 and 7, which came a bit unexpectedly for me as a reader.*

This description of the sections, including the seed convergence and starting point convergence was added both to the abstract and the end of Section 1.

At the end of the abstract - "The required number of random seeds for stochastic environmental conditions is considered to ensure that the sensitivities are due to the input parameters and not due to the seed. The required number of analysis points in the parameter space is identified so that the conclusions represent a global sensitivity. The results are specific to the platform and turbine, but the demonstrated approach can be applied widely to guide focus in design parameter uncertainty."

At the end of Section 1 this outline paragraph was added – "A demonstration of this is shown below. Section 2 describes the chosen FOWT system to analyze and the method of analysis including numerical models and post-processing calculations. Section 3 defines the studied input parameters and their possible ranges. Sections 4 and 5 describe the output loads and their resulting sensitivities. The discussed findings are dependent on two analysis parameters, and the convergence of the sensitivities with respect to these choices is displayed in Sections 6 and 7."

- *Line 103-107: This is redundant, repeating the lines in L128-L133. I recommend removing Lines 103-107 and simply referring to the more in-depth description in L180-L133.*

The mention of the IEC recommendation in Line 103-107 is meant to explain the choice of simulation length. The mention of the IEC recommendation in Line 128-133 is meant to explain the background of the required number of seeds. The second reference was shortened to remove redundant text and information.

"The IEC recommended minimum of six seeds may be higher depending on the specific device and environmental condition (IEC, 2019a)."

- *Line 134: Consider explicitly saying how many seeds were chosen and including a reference to the seed convergence in Section 6.*

A reference to the seed convergence section was added in the line above.

- *Line 146: "partial derivative type" -> "partial-derivative-like"*

This change was made.

- *Line 144 – 151: This entire paragraph is confusingly worded, in my opinion, and thus difficult to follow. What is capital delta? And how do you calculate u_iw,range from your inputs? Consider revising and clarifying.*

The paragraph was reworked, and some non-essential text was removed. Capital delta is the change in the input parameter (perturbation size), and is now defined. U_iw,range is the total range of the input parameter (10 x the delta for our analysis).

- *Line 156: You are mixing notations in this equation, including both an overbar and vertical lines for the mean operator. Consider revising.*

The vertical lines are meant to designate absolute value and the overbar is for the output with all nominal valued inputs. The definition of this last term was fixed in the sentence above.

- *Line 184: Here you could reference some of the existing literature FOW loads-sensitivity studies to motivate your choice of input parameters (or verify that your list is inclusive of previously analysed parameters).*

The referenced works were the main motivation for the selected parameters and relevant ranges.

- *Line 190: Consider reiterating here that the variation of the input parameters was 10% of the range – I forgot this because it was only mentioned once previously and thought you were varying the parameters in the entire range.*

The entire range is sampled due to the changing starting points, but the perturbation from the starting point is always ±10%.

- *Line 203: "Justifications" should not be capitalized*

This change was made.

- *Line 209: This is not expected, considering Fig. 6 in [1]. Consider reviewing after you update your fatigue-load calculations.*

Line 209 is discussing the range of uncertainty in platform mass; are not sure if the reviewer meant this to be for another line as the figure in the source is about damage distribution compared to wind speed.

- *Line 300: If you are going to present Figures 4 and 5 in the main text, then I think they deserve a more thorough treatment than what is given here. Consider expanding this paragraph to not only include descriptions of the plots but also analysis of (a) why the trends make sense and (b) the important conclusions of the work. If there are no important conclusions, consider placing them in an appendix.*

The discussion of the results shown in these two figures has been expanded.

"The ultimate values shown in Figure 4 are strongly stratified by wind condition for most outputs. This separation is largely influenced by the changes in the nominal output, Y_w_bar, added to the EE value as shown in Equation 3. The ultimate EE value for the blade loads of the root bending moment and tip deflection are highest for the near-rated condition, logically where thrust is expected to be highest. The tower, shaft, and nacelle ultimate EE values are highest for the above-rated condition. The only outputs without a strong wind speed stratification are the global system motion-related quantities of mooring line tensions, watch circle, and heel angle. These three outputs are later shown to be largely dominated by current velocity and system center of mass, both input parameters that are independent of wind speed. The fatigue-proxy EE values are much more consistent between wind conditions, which is interesting, considering that each is weighted by the probability of the wind speed. This means that the differences in load variation between wind speed are roughly matched by the differences in wind speed probability. The spread in the ultimate values is larger than for the fatigue-proxy values."

- *Figure 4 and 5: Consider using a solid edge but a transparent face for the bar charts, to better prevent the bar on top from obscuring the bars plotted beneath.*

These histograms have stacked bars for the three wind conditions, so the part that is visible is the entire bar. The word stacked has been added to the caption to make this clear.

- *Figure 5: Consider using a logarithmic y-scale to better highlight the differences in the tails of the histograms.*

A logarithmic y-scale was considered, but then the breakdown of the wind conditions (as described in the previous comment) does not make sense. This linear scale was kept to show the distribution of events between wind conditions.

- *Figure 6 and 7: I really like these plots. A lot of nice information in a very concise way. Well done.*

Thank you, we thought it was helpful to see the type of events all together.

- *Line 324: Section 6 comes very unexpectedly, and it was not immediately clear to me what the objective of the section was. Consider rewriting the section slightly and adding a brief introduction that explains that the previously presented sensitivities are depending on the number of seeds, and that this section will go into the convergence of the sensitivities based on seed.*

The starting sentence of the section was edited to better link back to the previous results section.

First sentence of Section 6 has been edited to this – "The presented results in Section 5 require a certain number of seeds for the stochastic irregular waves and turbulent wind environment. This required number is specific to a given platform, turbine, and environmental condition."

- *Line 331 – 338: This paragraph was confusing to me, especially because you are analysing plots presented in the appendix. If it's important enough to deserve treatment in the main text, then the plots to be discussed should be in the main text. If they are not relevant to the main text, then they should be removed. My opinion is that Figures 8 and 9 are the "meat" of this section, and thus Appendix B can be removed entirely.*

Figures 8 and 9 show the necessary information to conclude the convergence of the results. This was deemed the most important, but you are not able to see the source of the problem, the relative variability due to seed number compared to the variability due to the parameter perturbation. We feel this information is important enough to leave in an appendix. The mention that these results exist in the appendix was kept, but the discussion of the plots themselves was moved to the corresponding appendix.

- *Line 356: This section ends abruptly. What is the final conclusion of the section?*

The conclusion of the section comes after the discussion of the fatigue-proxy results. Both type of loads need to be converged so the choice is not discussed until both load types have been presented.

- *Line 360: You can relate your conclusions to Fig. 3 in [1], although that was for an onshore turbine.*

This comparison has been added.

"A 2023 study looking into the statistical uncertainty in blade bending moment fatigue as a function of seed numbers also found that more than the IEC suggested minimum of six ten-minute simulations were needed (Mozafari et al., 2023). The Danish led project looked at a fixed onshore 10 MW turbine, and found that 50 seeds were needed for acceptable accuracy, and that more seeds were needed for lower wind speeds (Mozafari et al., 2023)."

- *Line 385: Why does adding more Sobol points increase the relative influence of sigma_u on UEE?*

The previous two paragraphs discuss how when the number of starting points is low, the unique results at a specific starting point skew the conclusions, and that the order of the starting points is fixed to follow

the Sobol sequence. It seems that the point at the start of the Sobol sequence happens to result in a relatively low sensitivity to sigma_u.

- *Appendix A: If you plot in log y-scale, you might not need this appendix and you can move this discussion to the main text, where I think it belongs.*

A logarithmic y-scale was considered, but then the breakdown of the wind conditions does not make sense. This linear scale was kept to show the distribution of events between wind conditions.

**Referee 2**

Major Comments:

*Many of the input ranges considered are not related to \*design\* values of each variable, but rather related to temporal variations in these variables (see explanation from Sørum et al.)*

The goal was to address a wide range of sources of parameter variability, including changes due to time, location, and manufacturing. The ranges assessed for all input parameters are meant to cover the full range for the lifetime of a project. The cited paper by Sørum et al. brings up a good point for design variable ranges. This design variable uncertainty approach could be used in future work that is focusing on a design specific analysis. We believe that the current method is useful to assess the sensitivity to the input parameters. Mention of this choice for the variable ranges has been added in the first paragraph of Section 3.1.

It should be noted that the parameter ranges represent the range of possible values for a given condition, and not a range of uncertainty for the design value at the top of the range. This applies for wind and wave conditions and generally increases the size of the parameter range, increasing the relative sensitivity.

*The environmental conditions are not necessarily relevant for extreme loads, yet ultimate loads are considered*

We agree that the operating load cases presented in this study likely do not result in the most extreme ultimate loads for some outputs. We plan to look into other load cases such as parked / idling conditions with extreme environmental states in future work. Not only do we expect that we could see some higher ultimate loads, we expect that the relative importance of wave parameter uncertainty could be higher for these load cases.

*Furthermore, the standard deviation is used as a stand-in for damage. Although this may be justified because of the short simulations (which would likely cause problems with unclosed cycles), only including the standard deviation and probability of occurrence is not very convincing. Overall, the results can be used to comment on how responses vary for different inputs, but not really to discuss uncertainties in design.*

The response and update for this comment are the same as for comment 1 of Referee 1.

*The choice of platform/turbine can also be interpreted as a weakness in the study, as the turbine is starting to be outdated and the platform is extremely heavy. (Consider the displacement of 14000 tonnes, compared to IEA VolturnUS 15 MW of 20000 tonnes or INO-WINDMOOR 12 MW of 14000 tonnes).*

*Furthermore, the turbine controller is simply detuned (assuming that the controller has not been modified since OC4), which is an unrealistic strategy regarding power tracking for a modern turbine.*

The authors agree and expect that the results could change with a larger and lighter system. Future work is planned to look at larger turbines and floaters. The advantage to this platform is the large amount of previous analysis and documentation. The demonstrated approach is more important than the specific results, and this design is very well known so readers can understand the context. The OpenFAST model is also very robust, avoiding physical or numerical instabilities across a wide range of parameters.

Minor Comments:

- *P4: Please define wave stretching more clearly. I suspect that the authors mean that wave loads are integrated to the mean free surface (while wave stretching could also refer to specifically Wheeler stretching).*

The drag forces are only applied up to the mean free surface. This has been added to the text for clarity.

- *P5: I'm not sure that Robertson et al. 2018 is the correct original reference for the radial EE.*

The reference was only an example of the method's use for a similar analysis.

- *P6: "a local partial derivative type calculation" – maybe this could be rephrased as a local partial derivative approximation using finite differences?*

This term was updated following Reviewer 1's comment (partial derivative-like).

- *Is it impossible to combine tables 2 and 3 for space reasons?*

This was attempted, but the required text size became too small.

- *P10: The "maximum wave height" should be rephrased as the "maximum significant wave height" because the maximum wave height is a different quantity.*

This change was made.

- *P12: The choice of ranges should be discussed more clearly with respect to what would be design values. Even though the ranges for some parameters are taken from previous work, the reader would benefit from being able to compare the reasoning behind these choices and the additional parameters (new to this study) in greater detail.*

For the full set of input parameters, the ranges were chosen to represent the possible expected range, not strictly the uncertainty around a design value. Clarification of this has been added in Section 3.1.

- *P12: Capitalization error "Justification"*

This change was made.

- *P13: Over what time period or with what probability are the maximum significant wave heights determined? These statistics need to be clearly defined.*

Both the significant wave height and peak period were calculated based on a 20 minute period. This has been added to the text.

- *P13: Is the breaking wave limit given for 200 m water depth? What water depth is considered for the buoy data?*

The breaking wave limit shown in Figure 3 is based on the 200 m water depth. The data is an aggregate of many sites with a wide range of different water depths.

- *P13: "site" - "sites"*

This change was made.

- *Fig 3: Perhaps the unphysical wave conditions should be removed from the figure (or shaded over)?*

We believe there is some value in showing what data is omitted in the analysis.

- *P14: It is interesting that the current is included, as this effect can be quite important. However, I'm not sure that the load models account for this effect completely. Vortex shedding is not included in the hydrodynamic load models, and there is no discussion of how the current is assumed to affect the waves (in particular the wave period).*

This is correct that vortex shedding effects are not modeled for the mooring hydrodynamic forces, only Morison element forces are included. In HydroDyn the current is a simple superposition on the wave velocities and no adjustments are made to the dispersion relation.

Added to the first paragraph of Section 2.2 – "Steady current was applied as a simple superposition on the wave velocities. No vortex shedding forces associated with the steady current are modeled."

- *P14: For some platforms, current in the opposite direction from the wind could actually give a larger pitch angle, depending on the shape of the platform. That might not be the case here, though.*

This is an interesting point. We imagine that this could be true for a spar-type or other deeper draft platform. This is a good consideration to keep in mind for future work.

- *P15: Line 271 – subject-verb agreement*

This change was made.

- *P20: I'm not convinced that aerodynamic damping is the right explanation for the response to waves. Damping is important when resonance is involved. Maybe aerodynamic damping is important for the resonant responses in the tower that are excited by waves, but those are high frequencies where there is little wave energy. At the main wave frequencies, there are no natural frequencies (by design). I suspect that the massive platform and the choice of ranges in the inputs have a greater effect here. Similarly, P22, it is note that "wind turbulence has high energy at low frequencies" – where we also have resonance, which makes these variations particularly important.*

While the effects of damping are most evident in resonant frequency ranges (which do not align with wave frequencies for this platform), they still have an effect on the response at the wave frequency. Additional comments have been added to highlight the importance of the very large and stable platform. The

sentence mentioning damping has also been edited to add that the operating turbine also increases the relative sensitivity to the already important wind parameters.

"The DeepCWind semi-submersible is a relatively large platform for the turbine size, reducing sensitivity to wave loading. That said, it should be recalled that all wind conditions involved an operating wind turbine, which adds considerable damping and increases the sensitivity to wind parameters. It is possible that the influence of the wave parameters may be much stronger for idling load cases, which will be addressed in future work."

- *P27: I would distinguish between watch circle and device footprint. These are two different things, as the footprint on the seabed will reach to the anchors regardless.*

The text was changed to use the term watch circle.

**Community 1**

*Regarding Section 3.1, parameter ranges, I feel that the paper would benefit greatly from a discussion of the choice of range and values of the tower stiffness. Specifically, if the natural frequency and relation to 3p/blade passing frequency were included it would help greatly to understand how these results might transfer to different scale turbines/floating platforms.*

The tower stiffness range resulted in a tower frequency range of 0.27 – 0.37 Hz (±15% around the base 0.32 Hz). The full extent of this range falls above the 1P frequencies and below the 3P frequencies tested in the study.

Added in Section 3.1 – "The tower stiffness range results in a ±15% change in tower frequency. This full range still falls above the turbine 1p and below the turbine 3P frequencies tested."

*Additionally, while in reference Robertson et al. 2019a, the mooring stiffness seems to have been a significant parameter while in this study it has not been included - is there a reason why this was left out? It would be of particular interest for concepts with fibre-based mooring designs whose stiffness can vary through the lifetime.*

The stiffness of the catenary mooring system used in this model is driven by the mass per unit length. This was included as an input variable. If fiber sections were included in the mooring design, we agree this would be an interesting variable to study.

This added description of the mooring lines was added to the end of section 2.1 – "The three lines are all composed of a single type of chain."

---

## Author Response (AR2)

Dear referees,

Thank you again for the reviews of the paper and the helpful feedback. Referee 2 had a few additional comments for a second edit. Below find the comments and the corresponding changes.

Thank you,

Will, Jason, and Amy

**Referee 2**

Comments:

*The revisions have improved the manuscript, although my previous comments on the relevance still stand. The study examines how different responses vary for different inputs, but the results are very much dependent on the choice of ranges for the parameters. The range for the wind speed standard deviation, in particular, is quite enormous, and I don't think that this range of variation is really relevant for design. All that being said, the work is clear and believable, and the results make sense in light of the choices made.*

The following text was added to emphasize the importance of the parameter range to the conclusions you are able to draw.

Page 10 – Section 3.1 - "It should be noted that the parameter ranges represent the range of possible values for a wide range of conditions, and not a range of uncertainty for the design value. This applies for wind and wave conditions and generally increases the size of the parameter range, increasing the relative sensitivity. The sensitivity to a parameter is directly correlated to the parameter range, so the resulting relative sensitivities need to be understood in the context of the parameter range choices. The presented range selection informs about the possible variation in loads across a wide range of conditions."

The last sentence of the conclusion was also updated to highlight this point.

Page 27 – Section 8 - "The conclusions should be treated as unique to the individual platform and turbine, as well as the selected parameter ranges, and it is recommended that nonoperational load cases are also considered."

*I don't find an answer regarding the 1 minute transients in the response, however – how is this duration determined to be sufficient to eliminate transient responses when the natural periods are longer than 1 minute?*

The text in italics was added to clarify this point. It is good to highlight that the initial conditions for surge and pitch were adjusted to be close to the expected mean value.

Page 5 – Section 2.2 - "Each simulation was run for a 10-minute time series with a 1-minute transient removed from the results. *This transient period was selected based on time series of the nominal load case for each of the three conditions. The time of the transient period was reduced by using initial surge and pitch values near their expected mean values for each wind speed.*"

*Abstract: I would reword the last sentence as "The results are specific to the platform, turbine, and choice of parameter ranges, but the demonstrated approach can be applied widely to guide focus in design parameter uncertainty.*

This change in the abstract has been made. Additionally, the word 'design' was removed from the end of the sentence. In line with the first comment, the parameter ranges are not necessarily ranges in 'design' values, but instead the range of expected possible conditions.

"The results are specific to the platform and turbine, and choice of parameter ranges, but the demonstrated approach can be applied widely to guide focus in parameter uncertainty."